# The Cambridge Law Corpus:
# A Dataset for Legal AI Research

**Andreas Östling**[1]    **Holli Sargeant**[2]    **Huiyuan Xie**[2]    **Ludwig Bull**[3]
**Alexander Terenin**[2]    **Leif Jonsson**[4]    **Måns Magnusson**[1]    **Felix Steffek**[2]
[1]Uppsala University    [2]University of Cambridge    [3]CourtCorrect    [4]Sudden Impact AB

## Abstract

We introduce the Cambridge Law Corpus (CLC), a corpus for legal AI research. It consists of over 250 000 court cases from the UK. Most cases are from the 21st century, but the corpus includes cases as old as the 16th century. This paper presents the first release of the corpus, containing the raw text and meta-data. Together with the corpus, we provide annotations on case outcomes for 638 cases, done by legal experts. Using our annotated data, we have trained and evaluated case outcome extraction with GPT-3, GPT-4 and RoBERTa models to provide benchmarks. We include an extensive legal and ethical discussion to address the potentially sensitive nature of this material. As a consequence, the corpus will only be released for research purposes under certain restrictions.

## 1 Introduction

In recent years, transformer-based neural networks (Vaswani et al., 2017) have transformed the field of textual data analysis. These models have reached or surpassed human performance on many classical natural language processing tasks (Devlin et al., 2019; Liu et al., 2019). These recent developments have made it possible to train token prediction models at a larger scale than ever before, using extensive textual corpora gathered from sources such as social media, books, newspapers, web links from Reddit, and Wikipedia (Brown et al., 2020; Zhang et al., 2022). In late 2022, OpenAI released ChatGPT, a chatbot based on the GPT-3.5 language model, to the public. Just a few months later, OpenAI made the more powerful GPT-4 model available, both via ChatGPT and through an API (OpenAI, 2023). GPT-4 has shown promising results on a large variety of tests designed for humans, such as AP tests, the LSAT and the Uniform Bar Exam (Martínez, 2023).

*Legal artificial intelligence* is emerging as a rapidly-developing area of machine learning (Zhong et al., 2020), which focuses on problems such as case outcome prediction (O'Sullivan and Beel, 2019; Chalkidis et al., 2019), legal entity recognition (Leitner et al., 2019; Dozier et al., 2010), prediction of relevant previous cases or relevant statutes (Liu et al., 2015), contract reviews (Hendrycks et al., 2021), and more recently, passing legal exams (Choi et al., 2023; Katz et al., 2023). Using machine learning models to solve such tasks automatically presents a new way for citizens and businesses to interact with many aspects of the legal system.

The release of sizeable, specialised datasets has been pivotal to machine learning research progress. Most famously, Deng et al. (2009) introduced the ImageNet dataset, containing over 1 million images spanning 1000 classes, crucial for the early development of deep convolutional neural networks (LeCun et al., 2015). Similarly, large datasets such as the Google Book corpus (Davis, 2011) have played an important role in the development of transformers and the popularisation of bread-and-butter techniques like byte-pair encoding (Gage, 1994). Just as critically, in parallel with these large-scale

---

The Cambridge Law Corpus's project page, which includes example data and terms of use, can be found at: HTTPS://WWW.CST.CAM.AC.UK/RESEARCH/SRG/PROJECTS/LAW. DOI: 10.17863/CAM.100221.

37th Conference on Neural Information Processing Systems (NeurIPS 2023) Track on Datasets and Benchmarks.

general-interest resources, many other specialised datasets have proven fruitful for research (Rasmy et al., 2021; Deng et al., 2009; Deng, 2012; Lin et al., 2015). Therefore, high-quality datasets for legal artificial intelligence research are needed to facilitate research progress.

Legal language tends to be more specialised than other kinds of natural language. Legal terms and idioms, which are often difficult for a lay person to understand, have strong semantics and are often the keystone of legal reasoning (Nazarenko and Wyner, 2017; Dale, 2017). For example, the term *stare decisis* is a legal "term of art" that has a specific legal meaning. The translation from Latin is "to stand by things decided": this term refers to the legal principle of *precedent*, that is, that courts must adhere to previous judgments of the same court (or judgments of higher courts) while resolving a case with comparable facts. We discuss this example and related points in Section 2.1.

Specialised large language models, such as recent legal versions of the BERT model (Chalkidis et al., 2020; Masala et al., 2021; Zheng et al., 2021), have been shown to perform better when fine-tuned on legal corpora. Suitable training data is needed in order to make the development of such models possible. At present, large research-quality legal corpora are not as widely available as analogous resources in other areas of machine learning. There is a lack of general large legal corpora focused on the United Kingdom, especially in machine-readable formats and with relevant annotations to facilitate the development of methods for solving key machine-learning tasks.

Legal data involves unique challenges, including the need to comply with regulations such as the GDPR legislation and to appropriately address privacy and other ethical matters. These challenges also include the need to digitise cases whose presentation is not in any manner standardised, such as those that are hundreds of years old but which constitute a legal precedent and whose content defines the law in use today. Handling details such as these creates a need for maintaining an appropriate level of dataset quality.

This work introduces a corpus containing, in its present form, 258,146 legal cases from the UK, available for research. We release the corpus and annotations of case outcomes for 638 cases.

**Prior Work and Current State of Affairs**

Corpora containing legal decisions in English-language jurisdictions other than the UK are becoming increasingly available, with examples such as MultiLegalPile (Niklaus et al., 2023) and LEXTREME (Chalkidis et al., 2023) providing court decisions in the United States, Canada, and Europe. Notably, LeXFiles (Chalkidis et al., 2023) contains legislation and case law from six countries—including the UK, which is our focus—albeit at a limited scale. Complementing case judgments, corpora such as CUAD (Hendrycks et al., 2021) provide expert-annotated legal contracts.

The availability of legal judgment data varies by jurisdiction and is influenced by (i) the degree to which legal judgements are anonymised, (ii) the degree to which they are standardised, and (ii) the degree of privacy protection offered by the jurisdiction. For example, Hwang et al. (2022) have recently released a corpus of Korean legal judgments. These judgments are anonymised when they are written, in accordance with the Korean legal system, making such a release more straightforward compared to other jurisdictions which do not anonymise. Other examples of court decisions include, Xiao et al. (2018) who released a large corpus of Chinese criminal cases, and Poudyal et al. (2020), who published a corpus from the European Court of Human Rights.

Compared to these examples, the availability of UK data is more limited. In terms of the aforementioned factors, the UK (i) does not anonymise its legal decisions or (ii) present them in a standardised format while simultaneously offering (iii) comparatively strong privacy protections. Recently, the UK's National Archives have started a service making case law available. At present, however, this service is limited to cases decided after 2003, prohibits bulk download and does not allow computational analysis by its applicable license (The National Archives, 2023).

## 2 The Cambridge Law Corpus

The Cambridge Law Corpus (CLC) is a corpus containing legal cases, also called judgments or decisions, from the combined United Kingdom jurisdiction, and England and Wales. Courts from Scotland and Northern Ireland are currently excluded. Further details on content are available in Appendix A, and in this work's companion datasheet (Gebru et al., 2021). Before summarising the

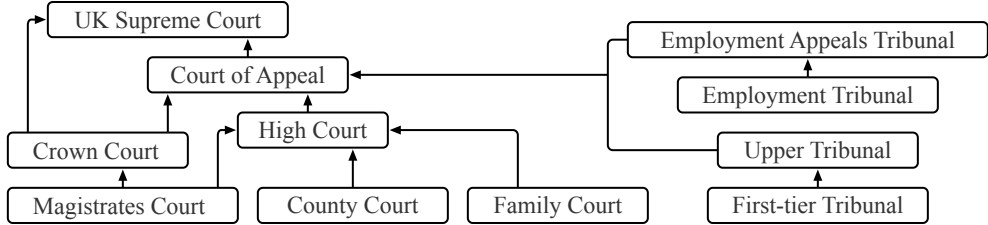

Figure 1: A simplified view of the UK court and tribunal structure (Judiciary UK, 2023).

corpus's content, we begin by reviewing some of the legal background needed to understand the material. Legal and ethical implications are addressed in Section 3.

## 2.1 The United Kingdom's Legal System

The UK does not have one single unified legal system—instead, it has three distinct jurisdictions: England and Wales, Scotland and Northern Ireland. There are exceptions to this rule, where some courts or tribunals hear cases from across the three jurisdictions. The common law of England and Wales is one of the major global legal traditions. It has been adopted or adapted in numerous countries worldwide.

Each jurisdiction is a common law jurisdiction— that is, authoritative law includes both legislation and court judgments. Legislation comprises Acts of Parliament (under the authority of the Crown) and regulations, that is, rules that govern how we must or must not behave and contain the consequences of our actions. Judges apply legislation and case law principles decided in previous cases to explain their decisions. Judges in different courts may decide cases with different principles across these three jurisdictions. Court judgments may be appealed to the highest court—the UK Supreme Court.

Court decisions can operate as an independent primary source of substantive law (case law), meaning that courts can legally justify their decisions by applying case law in the same way they apply legislation (Raz, 2009). Lawyers often call past cases *precedents*, and in many contexts, precedents at least influence the decisions of courts when relevantly similar disputes arise. In Anglo-American legal systems, this legal principle is known as *stare decisis*. Courts in the UK apply *stare decisis* relatively strictly.

The authority of the court will depend on the judgment court's place in the court hierarchy (see Figure 1). Precedent may have binding authority if the judgment has been upheld by a higher court, or persuasive authority if the judgment is from a lower court (Cross and Harris, 1991; Lewis, 2021). For example, the Supreme Court (formerly the House of Lords) is the highest court in the United Kingdom, and its decisions bind all courts in the jurisdiction (Lewis, 2021). This makes common law jurisdictions different from civil law jurisdictions. In the civil law jurisdiction, prior judicial decisions are not considered "law" (Merryman and Pérez-Perdomo, 2019).

A person or party may only be judged by the law and punished for a breach of the law (Dicey, 1915). Therefore, judges are not able to refuse to apply legislation; only Parliament can change the substantive rules in legislation. However, judges are able to interpret the appropriateness of law as it has evolved in light of the aims of previous lawmakers and may acknowledge that they make a policy-influenced decision.

Different courts deal with criminal and civil law (civil law cases are different to civil jurisdictions). Civil cases consider remedies to a party who has an enforceable legal right arising from tortious claims, contract disputes, family matters, company articles and similar. Criminal cases concern the law of crime and punishment where the Crown prosecutes an accused whom the police have investigated.

## 2.2 Corpus Content

The corpus consists of 258 146 cases, spanning 53 courts, with cases as old as the 16th century such as for instance *Walsingham's Case* (1573). Most cases are from the 20th and 21st centuries. Detailed figures on the development of the number of cases can be found in Appendix A. In total, these cases include approximately 5GB of legal text, consisting of roughly 0.8B tokens.

In the Cambridge Law Corpus, each case is stored as a single XML file by court and year. We chose XML in order to be able to annotate both whole cases and parts of cases in a structured, easy-to-use fashion, and in order to support many character encodings, comments, user-defined tags and namespaces. Appendix B shows an example of what a stored file might look like.

In addition to the case files, we also store additional relevant information in separate tables. For example, the categorical outcome of an individual case. In this way, the XML files mainly store the textual content of the case and the CSV files store additional features separately. This simplifies adding new information and features to the corpus. Further, the corpus comes with a Python library for quickly converting the XML files into formats commonly used in machine learning settings such as the Hugging Face DATASETS class.

The corpus contains both legal information and additional technical information. Each case contains a case header that contains general information on the case, such as the judge, the claimant and defendant, the date of the judgment, and similar. The case header also includes information on the court and the date of either the final hearing or judgment delivery.

In addition to the case header, the corpus also contains the case body, which contains the body text of the case. The case body contains complete sentences of legal text. Usually, it starts with a summary of the facts of the case, followed by arguments and a decision, although there is not necessarily any formal structure, and therefore the content of the body varies between cases and over time.

The case outcomes are stored separately from the cases and follow a hierarchical structure with aggregate outcomes, such as claimant wins or claimant loses, and more detailed case outcomes, such as damages awarded or permission to appeal granted. For details on definitions of case outcomes and a detailed list, see Appendix D.

Each case in the corpus is assigned a unique identifier (Cambridge Law Corpus identifier, CLC-ID). For this, we have created universally unique identifiers (UUID), which are identifiers that can guarantee uniqueness across space and time (Leach et al., 2005). In addition, a case's metadata can contain legal identifiers, which are used by the legal community to identify cases. These allow cases to be found in a manner mirroring their indexing in law reports. We also include neutral case citations, which were introduced in the United Kingdom in January 2001 to facilitate identifying a case in a manner independent from citations assigned by the commercial providers of the law reports. For example, the cases decided by the Civil Division of the England and Wales Court of Appeal are neutrally cited as follows: [year] EWCA Civ [number].

## 2.3 Corpus Creation and Curation Process

The original cases of the Cambridge Law Corpus were supplied by the legal technology company *CourtCorrect*[1] in raw form, including Microsoft Word and PDF files. The Word files were cleaned and transformed into an XML format. PDF files were converted to textual form via optical character recognition (OCR) using the *Tesseract* engine (Kay, 2007, v4.1.1)—an estimate of the OCR error rate is given in Appendix A. The resulting text files were then converted to the XML standard format of the corpus. The original documents are stored separately for quality control purposes.

Due to the size of the Cambridge Law Corpus, it is not feasible to annotate or curate the whole corpus manually. Instead, we use a process inspired by Voormann and Gut (2008), whose principles can be summarised as follows:

1. Replace sequential corpus creation phases with an encompassing, cyclic process model and employ a query-driven approach.

2. Recognise general error potentials and take measurements for immediate modifications in each part of the corpus creation process.

3. Minimal effort principle: slim annotation schema and little upfront annotation.

These principles have inspired our corpus creation and curation process in two ways. First, our process focused on improving the corpus in many small iterative steps. These include adding new annotations, new metadata, new cases and correcting identified errors, such as OCR errors. Second, we adopt a release model consisting of many small corpus releases, following the general ideas of

---

[1] HTTPS://COURTCORRECT.COM.

semantic versioning (Preston-Werner, 2013). We treat both the actual content of the corpus—the XML and CSV files—and the accompanying Python library as two complementary parts of the corpus's application programming interface (API).

These two principles have multiple benefits both for us and for the users of the corpus. First, rapid releases keep the corpus up to date in that new releases are made as soon as corrections or additions have been made to the corpus. This also encourages users to report errors in the data back to us in a structured way. Second, semantic versioning helps communicate the effect of the release to the users of the corpus and when to expect a change in the corpus API. Changes to the corpus are quality controlled through a random sample of the edits made.

## 2.4 Case Outcome Annotations

To enable researchers to study case outcomes, we added manual annotations tagging the tokens which contain the outcomes for a subset of cases. A case's outcome describes which party or parties are (partly) successful with their application(s) and which parties are not (partly) successful. In addition, the outcome contains the legal consequence decided by the court or tribunal. For example, a party may be ordered to pay a certain sum of money to the other party, or a person may be sentenced to a certain number of months in prison. Legal research and legal practice are particularly interested in the outcome of court cases. For instance, before going to court, it is helpful for parties to understand the expected outcome. After courts have decided a case, parties, advisers and researchers may be interested in knowing whether the decision is correct.

While it is easy to identify the outcome in the text of some cases, other cases are more difficult to understand. Some judgments report the speeches of multiple judges and it may be difficult to identify the final decision of the collective. As there are no formal rules or conventions on the writing of court decisions, the words and sentences defining the outcome vary considerably. In some cases, readers may even struggle to understand which legal outcome the judge aimed to lay down.

We started the annotation process by annotating a stratified random sample (by court) of legal cases. The annotations were made by legal experts at the Faculty of Law of the University of Cambridge following annotation instructions that can be found in Appendix D. The annotation instructions distinguish between the general outcome (in particular, whether a party is successful in court) and the detailed case outcome (for example, what remedy one party owes the other party). Following the ideas of Voormann and Gut (2008), we continuously updated the schema and instructions during the annotation process as problems arose.

The annotation process was undertaken by four lawyers (all of whom have a law degree, and three of whom are qualified solicitors and have graduate degrees in law) and one further legal expert (a Professor at the University of Cambridge's Faculty of Law, who also has higher qualifications in law) overseeing the annotation process. The four annotators and the legal expert met after a certain number of cases had been annotated to discuss problems that had arisen in the meantime. The main challenge was to align the annotation practice of the four annotators and to deal with new kinds of detailed case outcomes arising from the cases. As the corpus contains cases from various courts and tribunals as well as various areas of law, very specific detailed outcomes needed to be integrated into the annotation guidelines to capture such variability. While there is some research into formalising court outcomes, there is currently no accepted standard taxonomy of court outcomes the annotations could have referred to. Against this background, another contribution of this research is the first set of standardised annotation guidelines for UK case outcomes.

## 3 Legal and Ethical Considerations

The legality and ethics of collecting, processing and releasing the corpus is of paramount importance. Therefore, we have considered relevant safeguards to ensure legal compliance and the ethical design of this project. We now discuss these matters in more detail.

UK legislation and court decisions are subject to Crown copyright and licensed for use under the Open Government Licence. The Open Government Licence grants a worldwide, royalty-free, perpetual and non-exclusive licence (The National Archives, 2014).

The Data Protection Act 2018 (*DPA*) complements the UK implementation of the European Union's General Data Protection Regulation (*GDPR*), here referred to as the UK GDPR. Compliance with the DPA and UK GDPR has been the basis of the legality of this project as affirmed by the ethics approval from the University of Cambridge.

There are limitations on when personal data can be collected, processed and stored. The personal data in this corpus was not collected directly from data subjects and was only undertaken for research purposes in the public interest. Both these circumstances offer exemptions from obligations in the UK GDPR (DPA sch 2, pt 6, paras 26, 27; UK Information Commissioner's Office, 2022).

Given the practically impossible and disproportionate task of informing all individuals mentioned in this corpus and that these cases are publicly available and being processed exclusively for research purposes, makes this corpus exempt from notification requirements (DPA sch 2, pt 6, paras 26, 27; GDPR art 14(5)). Further, research in the public interest is privileged as regards secondary processing and processing of sensitive categories of data restrictions (GDPR art 6, rec 47, 157, 159). In particular, this aids the protection of the integrity of research datasets.

There are still important safeguards and considerations relevant to processing data of this sensitive nature. We apply safeguards in compliance with legal and ethical requirements and ensure that:

- Appropriate technical and organisational safeguards exist to protect personal data (DPA s 19; GDPR art 89)
- Processing will not result in measures being taken in respect of individuals and no automated decision-making takes place (DPA s 14; GDPR art 22).
- There is no likelihood of substantial damage or distress to individuals from the processing.
- Users who access the corpus must agree to comply with the DPA and the UK GDPR in addition to any local jurisdiction.
- Any individual may request the removal of a case or certain information which will be immediately removed (GDPR art 17; DPA s 47).
- The corpus will not pose any risks to people's rights, freedoms or legitimate interests (DPA s 19; GDPR art 89).
- Access to the corpus will be restricted to researchers based at a university or other research institution whose Faculty Dean (or equivalent authority) confirms, inter alia, that ethical clearance for the research envisaged has been received.
- Researchers using the corpus must agree to not undertake research that identifies natural persons, legal persons or similar entities. They must also guarantee they will remove any data if requested to do so.

We have considered the potential impact of processing such data. Given the public availability of all cases in the dataset in other repositories, the principle of open justice, the prohibition of research identifying individuals, the requirement of ethical clearance and our mechanisms for the erasure of data, we believe there is unlikely to be any substantial damage to individuals.

In the UK, court cases are not anonymised. Parties should, therefore, expect to be named in judgments because courts uphold the principle of open justice, promote the rule of law and ensure public confidence in the legal system (Judiciary UK, 2022). However, the court will anonymise a party if the non-disclosure is necessary to secure the proper administration of justice and to protect the interests of that party (CPR 39.2(4)). This decision includes weighing up the interests and rights of a party against the need for open justice. For example, individual names in asylum or other international protection claims are often anonymised by the court (Judiciary UK, 2022). Also, legislation sometimes prohibits naming certain individuals, such as victims of a sexual offence or trafficking (SOAA s 1, 2), or children subject to family law proceedings (CA s 97(2)).

While considering the ethics of this project, we examined the approaches of other jurisdictions to evaluate any relevant ethical and comparative legal considerations. Drawing from a jurisdiction with the opposite approach to the UK, in 2019, France amended its Justice Reform Act to prohibit the publication of judge analytics (Law 2019-22). The French restriction is based on express rules and a legal culture that varies considerably from the English common law system. France is a civil law system with different rules on how judges engage in cases, the binding nature of decisions and anonymisation (Cornu, 2014). For example, in the French legal system, individuals' names are

| Model | Acc | F1 | WER | BERTScore | BLEU | ROUGE$_L$ |
|---|---|---|---|---|---|---|
| End-to-end RoBERTa | 0.997 | 0.185 | 0.536 | 0.207 | 0.010 | 0.185 |
| RoBERTa pipeline | 0.739 | 0.012 | 0.776 | 0.030 | 0.015 | 0.028 |
| GPT-3.5 | - | - | 4.281 | 0.788 | 0.217 | 0.300 |
| GPT-4 | - | - | 3.396 | 0.840 | 0.282 | 0.517 |

Table 1: Evaluation results for the case outcome extraction task. The RoBERTa-based models are fine-tuned on the CLC annotations, while the GPT-based models are zero-shot—that is, they are evaluated without fine-tuning. We calculate accuracy and F1 scores for the RoBERTa-based models, and word error rate (WER), BERTScore, BLEU and ROUGE$_L$ for all baseline models.

usually anonymised. The French Justice Reform Act extends this restriction such that the "identity data of judges and court clerks may not be reused for the purpose or effect of evaluating, analysing, comparing or predicting their real or supposed professional practices" (Tashea, 2019). We have not identified any similar rules in the UK.

At present, the UK GDPR protects against autonomous decision-making that produces a legal or substantive effect on any individual. Researchers have to agree to comply with these and all GDPR rules to access the corpus, as well as any relevant obligations in their local jurisdiction. Taking a cautious approach, access to our corpus requires the researchers' promise not to use it for research that identifies natural persons, legal persons and similar entities.

We have considered the importance of using this corpus in a way that is in the public interest. Ethics approval has helped shape our safeguards and consider relevant risks. The emerging field of AI ethics evidences a growing interest in ensuring fair and just outcomes, in addition to legal compliance, as more personal data is used in algorithmic processes. Based on our legal and ethical evaluation of the project, it does not raise material risks to individuals' rights, freedoms or interests. Our safeguards also act to protect against any risks in further use of the corpus. The purpose, design and method of the corpus avoids such risks and prioritises positive outcomes that may be achieved through improving access to and research of case law. We will evaluate the requirements for access to the dataset on an ongoing basis to potentially further widen access, in particular, if the national and international legal framework becomes more permissive.

## 4 Experiments

As part of the curation of the Cambridge Law Corpus, building upon the corpus creation process and the additional annotations, we have undertaken a thorough exploration of two research tasks using English judgments, namely case outcome extraction and topic model analysis. These tasks offer a comprehensive range of possible use cases for leveraging the CLC in legal analysis.

### 4.1 Case Outcome Extraction

Court rulings can be complex and exceedingly lengthy, making them less accessible to non-legal professionals. To explore how the corpus can potentially improve such accessibility, we formulated a text classification task to identify text segments within a court decision that explicitly indicate the judges' decision regarding the final case outcome. This is done in two stages: we first classify whether or not each token in the case concerns the outcome, and then, given the result of this classification, we extract what the actual outcome is. A full, detailed description of this task is given in Appendix C.

Quality annotations on case outcomes are essential for this task in order to produce and evaluate output that is useful for legal analysis. We further provide manually annotated labels regarding case outcomes, including general case outcomes and detailed outcomes. The fine-grained annotations enable the extraction of tokens that enclose case outcomes from variable lengths of legal ruling texts. An example of the case outcome annotations is illustrated in Appendix D.

We sampled a total number of 638 cases (173,654 sentences) within the CLC to be manually annotated by legal experts, which are split into train/validation/test sets consisting of 388/100/150 cases, respectively. The number of cases and segments for each data split is summarised in Table 5.

| GPT-3.5-turbo | GPT-4 | Annotator Reference |
|---|---|---|
| Lord Justice: I also agree. | *I would dismiss these appeals.* | I would dismiss these appeals. |
| I would answer the issue: *The Claimant did comply with the service requirements of the Share Purchase Agreement.* I will hear submissions as to the next steps to be taken in relation to this claim, and as to whether it is appropriate to embody the answers to the preliminary issues (disputed and undisputed) in formal declarations. | *The Claimant did comply with the service requirements of the Share Purchase Agreement.* | The Claimant did comply with the service requirements of the Share Purchase Agreement. |
| *For the reasons I have given on each of the questions charterers are unsuccessful in their appeal.* Here too it is unnecessary to address procedural questions and I consider it undesirable to seek to do so. It follows from my analysis above that charterers' appeal on questions 2 and 3 fails on the merits. | *For the reasons I have given on each of the questions charterers are unsuccessful in their appeal.* | For the reasons I have given on each of the questions charterers are unsuccessful in their appeal. |

Table 2: Examples of outputs from GPT-3.5-turbo and GPT-4 and annotated references. Text highlighted in *green italics* denotes model output content that correctly correspond to annotations.

To extract case outcomes, we explored a number of approaches: (i) end-to-end RoBERTa, which uses the RoBERTa model to directly predict tokens containing case outcomes, (ii) a two-step RoBERTa pipeline, where we first extract the true sentences containing case outcome, then apply token-level RoBERTa classification on only those sentences, (iii) zero-shot GPT-3.5-turbo, which asks OpenAI's GPT-3.5-turbo language model to extract the exact text containing the case outcome from its full text, and (iv) zero-shot GPT-4, similar to (iii) except that we use the larger, more-powerful GPT-4 model. Note that the RoBERTa models are fine-tuned using our annotations, whereas the GPT models are not. Further details on our setup can be found in Appendix F.

Table 1 presents the evaluation results of the four baselines. Both RoBERTa-based models obtain relatively high accuracies, but comparatively low F1 scores, with end-to-end RoBERTa favouring a tradeoff more in the direction of higher accuracy. Both GPT models have a word error rate higher than 3, indicating that they produce too much text. Based on annotator feedback, this can be preferable, since it can be easier to work with output which contains the correct case outcome together with irrelevant sentences, compared to output that does not capture the outcome in enough detail.

Since, in real-world court ruling statements, the case outcome is described with very few words in relation to the total number of words in a case, the case outcome extraction task involves a large class imbalance. Moreover, the variability in how judgments are presented is very high: an ablation study, which uses different train-test splits to understand some of the variability, is given in Appendix C. Given these properties, models can simply learn to classify every sentence as not containing case outcome information, thus resulting in a very high accuracy but a very low F1 score. This may explain why, compared to the GPT models, the RoBERTa-based models exhibit lower scores in metrics such as BERTScore, BLEU, and ROUGE$_L$. This imbalance is an inherent challenge in the case outcome extraction task.

GPT-based models do not directly perform classification for individual tokens, but instead, given a prompt, produce a text in natural language. We constructed a prompt for case outcome extraction as follows. First, we set the *system message* so that it asks the model to be a legal research assistant. Then, in the main prompt, we asked the model to carefully read and then quote back exact text from the case text given, with certain extra formatting details. Full details on prompts can be found in Appendix F.

To evaluate the natural-language output produced by the GPT-based models, we use the word error rate (Morris et al., 2004) evaluation metric to measure how close the generated text is to the text segmented by annotators. Examples of the models' outputs and our gold standard annotations are demonstrated in Table 2, showing strong performance, especially for the larger GPT-4 model. Our results therefore suggest that GPT-based models can produce output which mirrors human annotators.

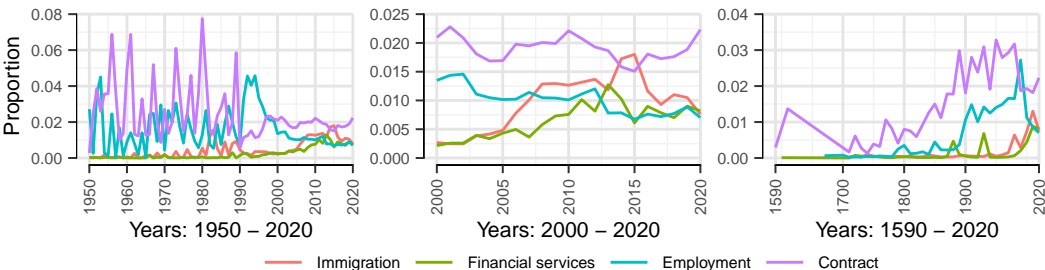

Figure 2: Proportion of words in documents belonging to the listed topics. A word can belong to more than one topic. Left: Aggregated to a one-year period spanning 1950-2020. Centre: Aggregated to a one-year period spanning 2000-2020. Right: Aggregated to a ten-year period spanning 1573-2020.

These examples illustrate that CLC can be a useful resource for benchmarking, fine-tuning and potentially pre-training large language models for legal tasks.

## 4.2 Topic Model Analysis

The areas of law courts deal with change over time, reflecting social, technological, economic and other societal changes. Understanding the areas of law courts are dealing with can enable one to understand, for instance, where conflicts in society arise and what resources and competencies are needed in the administration of justice. Practically, understanding the areas and numbers of cases going to court supports court managers in creating the necessary infrastructure. At the same time, a better understanding of the types of cases decided by the judges helps to identify areas where legal remedies offered in the court system are not effective. This would be the case, for example, if legal conflicts relevant to a large number of citizens and businesses are not reflected in court cases.

To illustrate a possible use case for how the CLC can be used to follow and analyse changes in topics of law over time, we ran a latent Dirichlet allocation (Blei et al., 2003) topic model using the parallel partially-collapsed Gibbs sampling algorithm (Magnusson et al., 2017; Terenin et al., 2019). We trained for 1200 iterations with hyper-parameters $\alpha = 0.1$, $\beta = 0.1$, $K = 100$ topics, a rare word threshold of 20 occurrences, and the default stop word list provided by Mallet (McCallum, 2002). The model reached a log-likelihood of $-3.0832 \times 10^9$ after 1200 iterations, at which point we obtained a sample of topic indicators from its output.

To analyse the topics over time, we aggregated the topic indicators produced on a per-year basis, which were then labelled by legal scholars of the Faculty of Law at the University of Cambridge. Figure 2 shows the development of four areas of law distinguishing three time periods: 1573-2020, 1950-2020, and 2000-2020. The top word tokens for these topics can be found in Appendix E.

From these results, we see that the *contract* topic is generally relevant at all times. This is expected, as a contract is a fundamental legal mechanism which allows natural and legal persons to privately order their cooperation. Interestingly, the relevance of contractual disputes in the courts drops somewhat from around 1990. This may reflect that businesses and consumers increasingly turn to other mechanisms than court litigation, such as alternative dispute resolution and in-house solutions to settle contractual conflicts.

The *financial services* topic, on the other hand, was hardly present in the courts until the 1990s. Its growing relevance in the courts may be caused by both increased public financial investment and the introduction of legal rules protecting consumers in financial services. The more recent drop in cases in the courts may also be a result of the rise of alternative dispute resolution, in particular, the dispute resolution services offered by the Financial Ombudsman Service.

One can similarly map rises and falls in the *immigration* and *employment* topics to historical events in the respective time periods. In total, this short analysis shows the potential of the CLC's use not only for legal AI, but also for legal research using computational methods.

As a final note, the cases available within the CLC are those cases that are available from the courts. In the UK, not all judgments are consistently published or otherwise made available. Courts have some discretion as to what to publish, may decide whether or not to publish in a manner that is not

fully transparent and may change their practice over time. This introduces a selection bias into the corpus: we provide additional details and discussion on this in Appendix A. From this viewpoint, the topic model analysis presented exemplifies a general way to determine what cases are available within the corpus.

## 5    Conclusion

We present the Cambridge Law Corpus, a collection of over 250 000 cases. We provide the corpus in an easy-to-work-with format, collected and converted from multiple sources and release the corpus using semantic versioning to simplify continuous improvement of the corpus. Due to the sensitive nature of the material, great care has been and will be taken in the ethical management and distribution of the corpus. We also provide a discussion on how to legally and ethically treat the data. Finally, we give two examples of how to use the corpus for research both in legal AI and legal research, case outcome extraction and topic model analysis, and provide a first benchmark on case outcome extraction.

## Acknowledgements

The work on the corpus is part of the UK Economic and Social Research Council (ESRC) and JST (Japan Science and Technology Agency) funded project on *Legal Systems and Artificial Intelligence*. The support of the ESRC and JST is gratefully acknowledged. We are grateful to Narine Lalafaryan, Joana Ribeiro de Faria and Lucy Thomas for excellent research assistance. We thank the participants of the Faculty of Law Research Seminar at the University of Cambridge for their helpful comments.

## General References

D. M. Blei, A. Y. Ng, and M. I. Jordan. Latent Dirichlet Allocation. *Journal of Machine Learning Research*, 2003. Cited on page 9.

T. B. Brown, B. Mann, N. Ryder, M. Subbiah, J. Kaplan, P. Dhariwal, A. Neelakantan, P. Shyam, G. Sastry, A. Askell, S. Agarwal, A. Herbert-Voss, G. Krueger, T. Henighan, R. Child, A. Ramesh, D. M. Ziegler, J. Wu, C. Winter, C. Hesse, M. Chen, E. Sigler, M. Litwin, S. Gray, B. Chess, J. Clark, C. Berner, S. McCandlish, A. Radford, I. Sutskever, and D. Amodei. Language Models are Few-shot Learners. In *Advances in Neural Information Processing Systems*, 2020. Cited on page 1.

N. Byrom. AI Risks Deepening Unequal Access to Legal Information. *Financial Times*, July 2023. URL: HTTPS://WWW.FT.COM/CONTENT/2ABA82C0-A24B-4B5F-82D9-EED72D2B1011. Cited on page 14.

I. Chalkidis, I. Androutsopoulos, and N. Aletras. Neural Legal Judgment Prediction in English. In *Association for Computational Linguistics*, 2019. Cited on page 1.

I. Chalkidis, M. Fergadiotis, P. Malakasiotis, N. Aletras, and I. Androutsopoulos. LEGAL-BERT: The Muppets Straight out of Law School. In *Conference on Empirical Methods in Natural Language Processing*, 2020. Cited on page 2.

I. Chalkidis, N. Garneau, C. Goanta, D. M. Katz, and A. Søgaard. LeXFiles and LegalLAMA: Facilitating English Multinational Legal Language Model Development. In *Association for Computational Linguistics*, 2023. Cited on page 2.

J. Choi, K. Hickman, A. Monahan, and D. Schwarcz. ChatGPT Goes to Law School. *Journal of Legal Education (Forthcoming)*, 2023. Cited on page 1.

G. Cornu. *Vocabulaire juridique*. Presses universitaires de France, 2014. Cited on page 6.

R. Cross and J. W. Harris. *Precedent in English Law*. Clarendon Press, 1991. Cited on page 3.

K. Dale. Legal Corpus Linguistics: Gambling to Gaming Language Powers and Probabilities Notes. *UNLV Gaming Law Journal*, 8(2):233–252, 2017. Cited on page 2.

M. Davis. Google Books (American English) Corpus, 2011. URL: HTTPS://VARIENG.HELSINKI.FI/CORD/CORPORA/GOOGLEBOOKS/. Cited on page 1.

J. Deng, W. Dong, R. Socher, L.-J. Li, K. Li, and L. Fei-Fei. Imagenet: A Large-scale Hierarchical Image Database. In *Computer Vision and Pattern Recognition*, 2009. Cited on pages 1, 2.

L. Deng. The MNIST Database of Handwritten Digit Images for Machine Learning Research. *IEEE Signal Processing Magazine*, 2012. Cited on page 2.

J. Devlin, M. W. Chang, K. Lee, and K. Toutanova. BERT: Pre-training of Deep Bidirectional Transformers for Language Understanding. In *Association for Computational Linguistics – Human Language Technologies*, 2019. Cited on pages 1, 19.

A. V. Dicey. *Introduction to the Study of the Law of the Constitution*. Macmillan, 1915. Cited on page 3.

C. Dozier, R. Kondadadi, M. Light, A. Vachher, S. Veeramachaneni, and R. Wudali. Named Entity Recognition and Resolution in Legal Text. In *Semantic Processing of Legal Texts*, 2010. Cited on page 1.

P. Gage. A New Algorithm for Data Compression. *C Users Journal*, 1994. Cited on page 1.

T. Gebru, J. Morgenstern, B. Vecchione, J. W. Vaughan, H. Wallach, H. D. Iii, and K. Crawford. Datasheets for Datasets. *Communications of the ACM*, 2021. Cited on page 2.

D. Hendrycks, C. Burns, A. Chen, and S. Ball. CUAD: An Expert-Annotated NLP Dataset for Legal Contract Review. In *Advances in Neural Information Processing Systems*, 2021. Cited on pages 1, 2.

W. Hwang, D. Lee, K. Cho, H. Lee, and M. Seo. A Multi-task Benchmark for Korean Legal Language Understanding and Judgement Prediction. *Advances in Neural Information Processing Systems*, 2022. Cited on page 2.

Judiciary UK. Practice Guidance – Anonymisation of Parties to Asylum and Immigration Cases in the Court of Appeal, 2022. Cited on page 6.

Judiciary UK. Structure of Courts and Tribunals System, 2023. Cited on page 3.

D. M. Katz, M. J. Bommarito, S. Gao, and P. Arredondo. GPT-4 Passes the Bar Exam. *SSRN Preprint*, 2023. Cited on page 1.

A. Kay. Tesseract: An Open-Source Optical Character Recognition Engine. *Linux Journal*, 2007. Cited on page 4.

P. J. Leach, R. Salz, and M. H. Mealling. A Universally Unique IDentifier (UUID) URN Namespace, 2005. URL: HTTPS://WWW.IETF.ORG/RFC/RFC4122.TXT. Cited on page 4.

Y. LeCun, Y. Bengio, and G. Hinton. Deep Learning. *Nature*, 2015. Cited on page 1.

E. Leitner, G. Rehm, and J. Moreno-Schneider. Fine-Grained Named Entity Recognition in Legal Documents. In *International Conference on Semantic Systems*, 2019. Cited on page 1.

S. Lewis. Precedent and the Rule of Law. *Oxford Journal of Legal Studies*, 41(4):873–898, 2021. Cited on page 3.

T.-Y. Lin, M. Maire, S. Belongie, L. Bourdev, R. Girshick, J. Hays, P. Perona, D. Ramanan, C. L. Zitnick, and P. Dollár. Microsoft COCO: Common Objects in Context. In *European Conference on Computer Vision*, 2015. Cited on page 2.

Y.-H. Liu, Y.-L. Chen, and W.-L. Ho. Predicting Associated Statutes for Legal Problems. *Information Processing & Management*, 2015. Cited on page 1.

Y. Liu, M. Ott, N. Goyal, J. Du, M. Joshi, D. Chen, O. Levy, M. Lewis, L. Zettlemoyer, and V. Stoyanov. RoBERTa: A Robustly Optimized BERT Pretraining Approach, 2019. Cited on pages 1, 19.

M. Magnusson, L. Jonsson, M. Villani, and D. Broman. Sparse Partially Collapsed MCMC for Parallel Inference in Topic Models. *Journal of Computational and Graphical Statistics*, 2017. Cited on page 9.

E. Martínez. Re-Evaluating GPT-4's Bar Exam Performance. *SSRN Preprint*, 2023. Cited on page 1.

M. Masala, R. C. A. Iacob, A. S. Uban, M. Cidota, H. Velicu, T. Rebedea, and M. Popescu. jurBERT: A Romanian BERT Model for Legal Judgement Prediction. In *Natural Legal Language Processing Workshop*, 2021. Cited on page 2.

A. K. McCallum. MALLET: A Machine Learning for Language Toolkit, 2002. URL: HTTPS://MIMNO.GITHUB.IO/MALLET/. Cited on page 9.

J. H. Merryman and R. Pérez-Perdomo. *The Civil Law Tradition: An Introduction to the Legal Systems of Europe and Latin America*. Stanford University Press, 2019. Cited on page 3.

A. Morris, V. Maier, and P. Green. From WER and RIL to MER and WIL: Improved Evaluation Measures for Connected Speech Recognition. In 2004. Cited on page 8.

A. Nazarenko and A. Wyner. Legal NLP Introduction. *Traitement Automatique des Langues*, 58(2):7–19, 2017. Cited on page 2.

J. Niklaus, V. Matoshi, M. Stürmer, I. Chalkidis, and D. E. Ho. MultiLegalPile: A 689GB Multilingual Legal Corpus. In *ICML Workshop on Data-centric Machine Learning Research*, 2023. Cited on page 2.

C. O'Sullivan and J. Beel. Predicting the Outcome of Judicial Decisions Made by the European Court of Human Rights. In *Irish Conference on Artificial Intelligence and Cognitive Science*, 2019. Cited on page 1.

OpenAI. GPT-4 Technical Report, 2023. Cited on page 1.

P. Poudyal, J. Savelka, A. Ieven, M. F. Moens, T. Goncalves, and P. Quaresma. ECHR: Legal Corpus for Argument Mining. In *Workshop on Argument Mining*, 2020. Cited on page 2.

T. Preston-Werner. Semantic Versioning, 2013. URL: HTTP://SEMVER.ORG/. Cited on page 5.

C. Raffel, N. Shazeer, A. Roberts, K. Lee, S. Narang, M. Matena, Y. Zhou, W. Li, and P. Liu. Exploring the Limits of Transfer Learning with a Unified Text-to-text Transformer, 2019. Cited on page 19.

L. Rasmy, Y. Xiang, Z. Xie, C. Tao, and D. Zhi. Med-BERT: Pretrained Contextualized Embeddings on Large-scale Structured Electronic Health Records for Disease Prediction. *npj Digital Medicine*, 2021. Cited on page 2.

J. Raz. *The Authority of Law: Essays on Law and Morality*. Oxford University Press, 2009. Cited on page 3.

C. Somers-Joce, D. Hoadley, and E. Nemsic. How Public is Public Law? The Current State of Open Access to Administrative Court Judgments. *Judicial Review*, 27(2):95–98, 2022. Cited on page 14.

J. Tashea. France Bans Publishing of Judicial Analytics and Prompts Criminal Penalty. American Bar Association Journal, 2019. Cited on page 7.

A. Terenin, M. Magnusson, L. Jonsson, and D. Draper. Pólya Urn Latent Dirichlet Allocation: A Doubly Sparse Massively Parallel Sampler. *IEEE Transactions on Pattern Analysis and Machine Intelligence*, 2019. Cited on page 9.

The National Archives. Find Case Law, 2023. URL: HTTPS://CASELAW.NATIONALARCHIVES.GOV.UK/. Cited on page 2.

The National Archives. Open Government Licence, 2014. URL: HTTPS://WWW.NATIONALARCHIVES.GOV.UK/DOC/OPEN-GOVERNMENT-LICENCE/VERSION/3/. Cited on page 5.

UK Information Commissioner's Office. Guide to Data Protection: Research Provisions, 2022. Cited on page 6.

A. Vaswani, N. Shazeer, N. Parmar, J. Uszkoreit, L. Jones, A. N. Gomez, L. Kaiser, and I. Polosukhin. Attention is All You Need. In *Advances in Neural Information Processing Systems*, 2017. Cited on page 1.

H. Voormann and U. Gut. Agile Corpus Creation. *Corpus Linguistics and Linguistic Theory*, 2008. Cited on pages 4, 5.

C. Xiao, H. Zhong, Z. Guo, C. Tu, Z. Liu, M. Sun, Y. Feng, X. Han, Z. Hu, H. Wang, and J. Xu. CAIL2018: A Large-scale Legal Dataset for Judgment Prediction. *arXiv Preprint 1807.02478v1*, 2018. Cited on page 2.

S. Zhang, S. Roller, N. Goyal, M. Artetxe, M. Chen, S. Chen, C. Dewan, M. Diab, X. Li, X. V. Lin, T. Mihaylov, M. Ott, S. Shleifer, K. Shuster, D. Simig, P. S. Koura, A. Sridhar, T. Wang, and L. Zettlemoyer. OPT: Open Pre-trained Transformer Language Models. Technical report, Meta AI, 2022. Cited on page 1.

L. Zheng, N. Guha, B. R. Anderson, P. Henderson, and D. E. Ho. When Does Pretraining Help? Assessing Self-supervised Learning for Law and the Casehold Dataset of 53,000+ Legal Holdings. In *International Conference on Artificial Intelligence and Law*, 2021. Cited on page 2.

H. Zhong, C. Xiao, C. Tu, T. Zhang, Z. Liu, and M. Sun. How Does NLP Benefit Legal System: A Summary of Legal Artificial Intelligence. In *Association for Computational Linguistics*, 2020. Cited on page 1.

## Legal References

Children Act 1989 (UK).

Civil Procedure Rules (UK).

Data Protection Act 2018 (UK).

Loi 2019-22 du 23 mars 2019 de programmation 2018-2022 et de réforme pour la justice [Law 2019-22 of 23 March 2019 on the Programming 2018-2022 and the reform for justice] 2019, Journal Officiel de la République Française (FR).

Regulation (EU) 2016/679 of the European Parliament and of the Council of 27 April 2016 on the protection of natural persons with regard to the processing of personal data and on the free movement of such data and repealing Directive 95/46/EC 2016, OJ L 119/1 (EU).

Sexual Offences Amendment Act 1992 (UK).

Walsingham's Case [1573] EWHC KB J99, [1573] 75 ER 805.

|      | Mean   | SE     |
|------|--------|--------|
| WER  | 0.0283 | 0.0097 |
| CER  | 0.0417 | 0.01   |

Table 3: Word error rate (WER) and character error rate (CER) for pages on which optical character recognition was performed.

# A    Detailed Information on Corpus Content

Table 4 gives the total number of cases for each court included in the Cambridge Law Corpus (CLC) and the abbreviations used in the following figures.

The corpus does not contain all UK cases. Not all judgments are consistently published or made otherwise available by the courts, even though access to court judgments is fundamental to the principle of open justice (discussed in Section 3) under the UK common law system. The Supreme Court and Privy Council generally publish their decisions comprehensively. However, lower court decisions may not always be captured by various sources and some discretion is held by judges and the judicial press office. Courts have discretion as to what judgments are published. For example, a study found that between 2015 and 2020 only 55% of 5,408 administrative court judgments identified by a commercial subscription service appeared on the website of the BAILII (Somers-Joce et al., 2022; Byrom, 2023). We did not explicitly select any cases in or out, except to exclude Scotland and Northern Ireland by jurisdiction, and include all cases that we were able to obtain.

We performed an analysis of the optical character recognition (OCR) error rate, by sampling 40 pages randomly from the pages OCRed by *Tesseract*, and then sampling 3 rows from each page for manual assessment—a two-stage cluster sampling design. We then compared the manually written sentences with the output from *Tesseract*. We calculate the mean word error rate (WER) and character error rate (CER) or the Levenshtein distance. Results can be found in Table 3.

| Court | Abbreviation | Cases |
|---|---|---|
| United Kingdom Competition Appeals Tribunal | CAT | 478 |
| England and Wales Court of Appeal (Civil Division) | EWCA-Civ | 18461 |
| England and Wales Court of Appeal (Criminal Division) | EWCA-Crim | 5787 |
| England and Wales County Court (Family) | EWCC-Fam | 111 |
| England and Wales Court of Protection | EWCOP | 531 |
| England and Wales Care Standards Tribunal | EWCST | 745 |
| England and Wales Family Court (High Court Judges) | EWFC-HCJ | 386 |
| England and Wales Family Court (other Judges) | EWFC-OJ | 873 |
| England and Wales High Court (Administrative Court) | EWHC-Admin | 11555 |
| England and Wales High Court (Admiralty Division) | EWHC-Admlty | 101 |
| England and Wales High Court (Chancery Division) | EWHC-Ch | 6623 |
| England and Wales High Court (Commercial Court) | EWHC-Comm | 2926 |
| Court of Common Pleas | EWHC-CP | 6 |
| England and Wales High Court (Exchequer Court) | EWHC-Exch | 11 |
| England and Wales High Court (Family Division) | EWHC-Fam | 2239 |
| Intellectual Property Enterprise Court | EWHC-IPEC | 170 |
| England and Wales High Court (King's Bench Division) | EWHC-KB | 34 |
| Mercantile Court | EWHC-Mercantile | 22 |
| England and Wales High Court (Patents Court) | EWHC-Patents | 642 |
| England and Wales High Court (Queen's Bench Division) | EWHC-QB | 4645 |
| England and Wales High Court (Technology and Construction Court) | EWHC-TCC | 1689 |
| England and Wales Land Registry Adjudicator | EWLandRA | 482 |
| England and Wales Lands Tribunal | EWLands | 570 |
| England and Wales Leasehold Valuation Tribunal | EWLVT | 18445 |
| England and Wales Patents County Court | EWPCC | 135 |
| English and Welsh Courts - Miscellaneous | Misc | 451 |
| Special Immigrations Appeals Commission | SIAC | 174 |
| United Kingdom Immigration and Asylum (AIT/IAC) Unreported Judgments | UKAITUR | 65034 |
| United Kingdom Employment Appeal Tribunal | UKEAT | 16002 |
| United Kingdom Employment Tribunal | UKET | 36772 |
| United Kingdom Financial Services and Markets Tribunals | UKFSM | 69 |
| First-tier Tribunal (General Regulatory Chamber) | UKFTT-GRC | 2311 |
| First-tier Tribunal (Health Education and Social Care Chamber) | UKFTT-HESC | 437 |
| First-tier Tribunal (Property Chamber) | UKFTT-PC | 7860 |
| First-tier Tribunal (Tax) | UKFTT-TC | 7684 |
| United Kingdom House of Lords | UKHL | 4929 |
| United Kingdom Asylum and Immigration Tribunal | UKIAT | 1352 |
| Information Commissioner's Office | UKICO | 13659 |
| United Kingdom Investigatory Powers Tribunal | UKIPTrib | 51 |
| United Kingdom Information Tribunal incl. National Security Appeals Panel | UKIT | 250 |
| The Judicial Committee of the Privy Council | UKPC | 10334 |
| United Kingdom Supreme Court | UKSC | 743 |
| United Kingdom Special Commissioners of Income Tax | UKSPC | 439 |
| UK Social Security and Child Support Commissioners' | UKSSCSC | 2467 |
| Upper Tribunal (Administrative Appeals Chamber) | UKUT-AAC | 3048 |
| Upper Tribunal (Immigration and Asylum Chamber) | UKUT-IAC | 811 |
| United Kingdom Upper Tribunal (Lands Chamber) | UKUT-LC | 1009 |
| United Kingdom Upper Tribunal (Tax and Chancery Chamber) | UKUT-TCC | 918 |
| United Kingdom VAT & Duties Tribunals | UKVAT | 2767 |
| United Kingdom VAT & Duties Tribunals (Customs) | UKVAT-Customs | 104 |
| United Kingdom VAT & Duties Tribunals (Excise) | UKVAT-Excise | 785 |
| United Kingdom VAT & Duties Tribunals (Insurance Premium Tax) | UKVAT-IPT | 7 |
| United Kingdom VAT & Duties Tribunals (Landfill Tax) | UKVAT-Landfill | 12 |
| | **Total** | **258146** |

Table 4: Courts included in the Cambridge Law Corpus

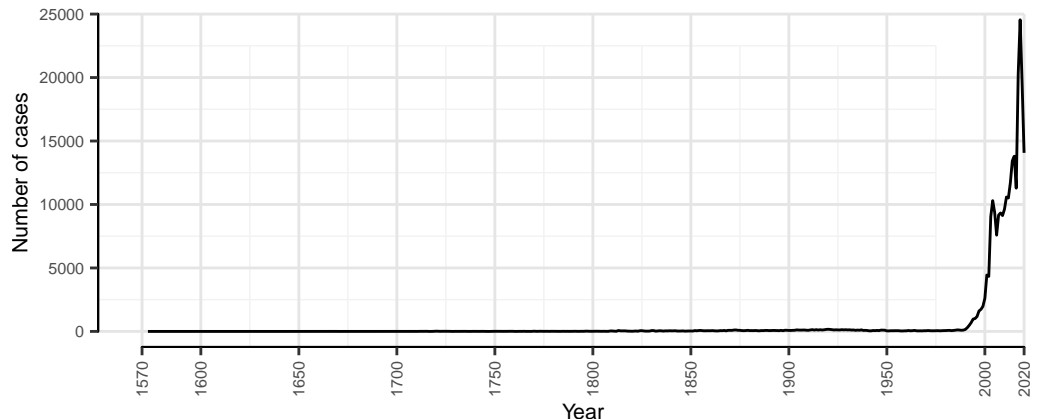

Figure 3: Number of cases per year.

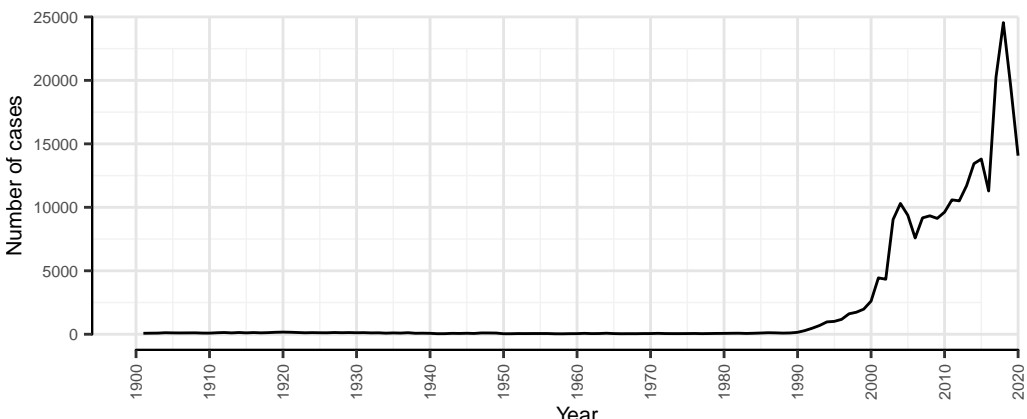

Figure 4: Number of cases per year, 1900 forwards.

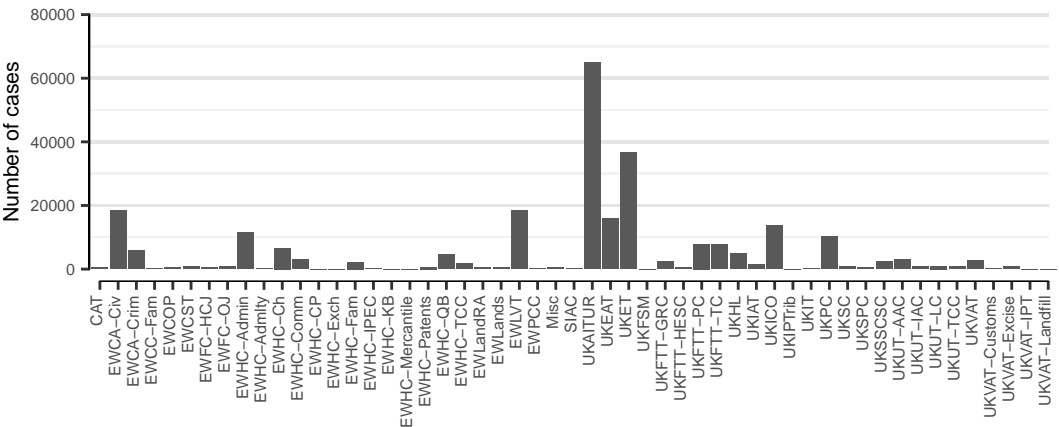

Figure 5: Number of cases per court.

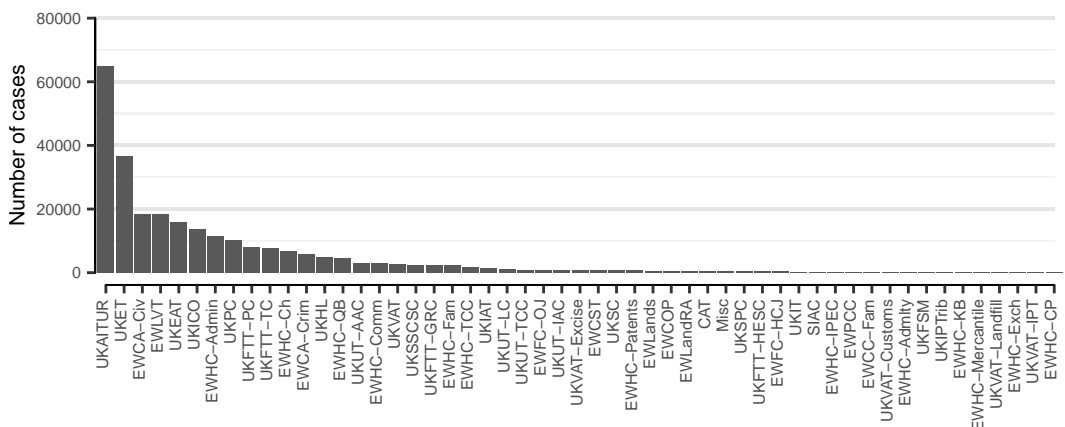

Figure 6: Number of cases per court in descending order.

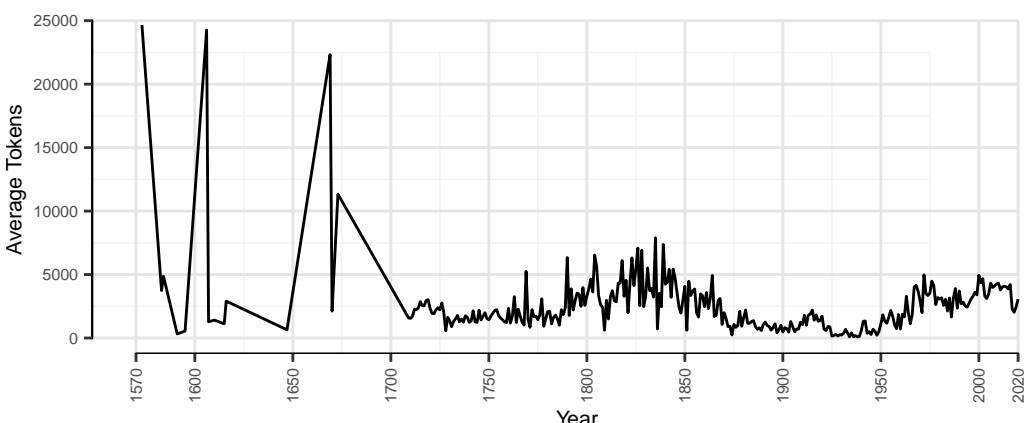

Figure 7: Number of tokens per case per year. Tokens are defined as separated by white space.

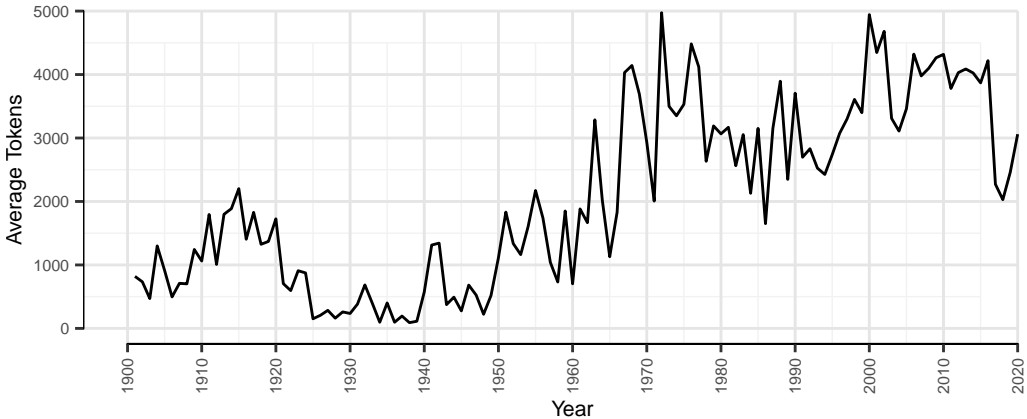

Figure 8: Number of tokens per case per year, from 1900 onward. Tokens are defined as separated by white space.

## B   Example XML case

```xml
<?xml version="1.0" encoding="UTF-8"?>
<basic_case_document
↪   xmlns="https://github.com/cambridge-ai-and-law-project/CambridgeLawCorpus"
↪   xmlns:xsi="http://www.w3.org/2001/XMLSchema-instance" xsi:schemaLocation="http
↪   s://github.com/cambridge-ai-and-law-project/CambridgeLawCorpus
↪   https://github.com/cambridge-ai-and-law-project/CambridgeLawCorpus/schemas/cas
↪   e_schema.xsd">
  <CLC-ID>5f916ee5-b3f9-4fb4-b044-1575765da226</CLC-ID>
  <court>England and Wales Court of Appeal (Civil Division)</court>
  <lower_court/>
  <last_date_hearing_delivery>2019-02-19</last_date_hearing_delivery>
  <neutral_citations>
    <case_neutral_citation>[2019] ICR 1155</case_neutral_citation>
    <case_neutral_citation>[2019] EWCA Civ 125</case_neutral_citation>
    <case_neutral_citation>[2019] IRLR 545</case_neutral_citation>
  </neutral_citations>
  <law_report_references>
  </law_report_references>
  <case_nr_of_year/>
  <judges>
  </judges>
  <claimants>
  </claimants>
  <defendants>
  </defendants>
  <solicitors>
  </solicitors>
  <barristers>
  </barristers>
  <parties>
  </parties>
  <case_text><![CDATA[
England and Wales Court of Appeal (Civil Division) Decisions

 <NAME>  v  <NAME> & <NAME> [2019] EWCA Civ 125 (19 February 2019)
[2019] ICR 1155,
[2019] EWCA Civ 125,
[2019] IRLR 545
Neutral Citation Number: [2019] EWCA Civ 125
Case No: A2/2017/0040
IN THE COURT OF APPEAL (CIVIL DIVISION) ON APPEAL FROM the Employment Appeal
↪   Tribunal Mitting J
Royal Courts of Justice Strand, London, WC2A 2LL
19/02/2019
B e f o r e :
LORD JUSTICE  <NAME> (Vice-President of the Court of Appeal (Civil Division)) LORD
↪   <NAME> and LORD JUSTICE  <NAME>
-------------------
Between:
Appellant
- and -
Respondents
-------------------

Ms <NAME> (instructed by  <NAME>) for the Appellant
Mr <NAME> QC and Mr  <NAME> (instructed by Waring & Co) for the Respondents
Hearing date: 3 October 2018

Lord Justice  <NAME>:
INTRODUCTION
The Claimants in these proceedings were employed, during the period with which we
↪   are concerned, by a company which has since been dissolved called
```

```
[...]

DISPOSAL I would dismiss the appeal

[...]

Lord <NAME>: I agree.
Lord Justice <NAME>: I also agree.
]]></case_text>
</basic_case_document>
```

## C Case Outcome Task Description

Below we provide additional details on the models used for the case outcome extraction task. Statistics for the data are given in Table 5, and an example annotated case outcome is given in Figure 9.

- End-to-end RoBERTa. In this setup, the RoBERTa (Liu et al., 2019) model is used to directly predict tokens that may contain case outcome information.

- Two-step RoBERTa pipeline. In this setup, we first extract sentences that explicitly contain case outcome information from those that do not. On sentences that contain case outcome information, we further perform token classification using RoBERTa, which is done similarly to the previous RoBERTa token classification setup. By dividing the problem into two levels of granularity—sentence-level and token-level—we can see whether explicitly enforcing an intermediate step helps achieve better end results for token classification.

- GPT-3.5-turbo in a zero-shot setting. Here we evaluate the performance of GPT-based models, which have been reported to achieve state-of-the-art results on multiple language-related benchmarks. In a zero-shot setting, we asked GPT-3.5-turbo to extract the exact text containing the case outcome when provided with the full case text. See Appendix F for further GPT-specific details.

- GPT-4 in a zero-shot setting. Here, we test the more-powerful GPT-4 model along similar lines as the GPT-3.5-turbo. GPT-4 is a multi-modal language model that allows multiple forms of input sources—including text and image—and accepts longer-form text as input compared to GPT-3.5-turbo. Note that supporting the latter format makes this model particularly suitable for the legal setting, which can include long cases.

**Ablation study on sentence classification step in the two-step RoBERTa pipeline** In this ablation study, we first report evaluation results obtained from the first component—that is, the sentence-level classifier—of the RoBERTa pipeline. After 20 epochs of training, the sentence-level RoBERTa classifier achieves an extraction accuracy of 82.3%.

We also explored an ablated specification where we sliced examples sequentially into train/validation/test splits rather than randomly shuffling all examples across courts. When cases are sequentially sliced from their original order in the CLC, cases in a particular split tend to belong to specific courts. For example, all test cases in this ablated setting are from the England and Wales High Court (EWHC) and the UK Supreme Court (UKSC), whilst cases in the other two splits are from different courts. As a result, cases in different splits are principally from different courts, as illustrated in Table 6. With this setting, studying extrapolation, we trained multiple sentence-level classifiers—BERT (Devlin et al., 2019) and T5 (Raffel et al., 2019)—to extract sentences that potentially contain case outcome

|            | Num. cases | Num. sentences | Num. sent. with outcome | Prevalence |
|------------|------------|----------------|-------------------------|------------|
| Training   | 388        | 99565          | 536                     | 0.54%      |
| Validation | 100        | 29208          | 126                     | 0.43%      |
| Testing    | 150        | 44881          | 208                     | 0.46%      |

Table 5: Data statistics for the annotations used in the case outcome extraction task.

| Split | Courts |
|---|---|
| Training | EWHC (Comm, Ch, Patents, Fam, QB, Admlty), EWCA (Civ) |
| Validation | EWHC (TCC), UKHL |
| Testing | UKHL, UKSC |

Table 6: Courts used in the sequentially sliced train/val/test splits.

information. Considering that in this ablated study, cases from each data split tend to belong to different courts compared to when the cases are randomly shuffled, extracting sentences that potentially indicate general case outcome in this case requires a stronger ability for the classifier models to generalise to unseen cases from other courts. The BERT model yields an extraction accuracy of 50.8%, whilst the T5 model yields an accuracy of 53.9%. These accuracy results are significantly lower than the extraction accuracy we obtained under the experiment setting where we shuffled all cases. This indicates that judge ruling statements vary significantly across English courts.

# D   Case Outcome Annotation Instructions

Below are the annotator guidelines for the annotation of general and detailed case outcomes.

## D.1   Case Outcome General

The case outcome is annotated in two steps. First the part of the case that contains the information of the case outcome is marked. Then this case outcome is classified both with a general and a detailed label describing the outcome of the case.

**How is the label used?**   The tag *case outcome general (label: CaseOutcomeGeneral)* is a document class with the categories:

[**CLAIMANT_WINS**]  Claimant wins

[**CLAIMANT_PARTLY_WINS**]  Claimant partly wins

[**CLAIMANT_LOSES**]  Claimant loses

[**OTHER_OUTCOME**]  Other outcome

[**NONE**]  Outcome cannot be identified

Please distinguish this general outcome tag from the detailed outcome tag below.

*Note*: Sentences such as where the judge says 'I agree' are not tagged as [CaseOutcomeGeneral] and not as [CaseOutcomeDetailed]. The reason for not tagging such sentences is that words such as 'I agree' do not indicate a specific legal consequence and only reference other text containing the legal consequence.

If there are multiple judges then [CaseOutcomeGeneral] is only tagged for the lead judge and not for other majority judges and not for minority judges.

In the circumstances of his forbearance and not getting £ < amount > besides the costs he has incurred all these years , it would not be appropriate further to pursue the periodical payments application. I would therefore dismiss this application for permission to appeal. CASE OUTCOME ⊗ LORD JUSTICE < NAME > : I agree .

Figure 9: An example of the case outcome annotations in the CLC.

**Definition**    The result of the case expressed as winning or losing from the claimant's perspective. There is also a category in case the result cannot be determined or if it is a case where the outcome cannot be described in terms of winning or losing (e.g., an evidence collection).

**Example**    If the claimant requires damages of 100 GPB and the court awards 100 GBP then it is: Claimant wins.

**What parts of the text do you want annotated, and what should be left alone?**    Please tag both general and detailed outcomes if possible.

### D.2    Case Outcome Detailed

**What is the tag called and how is it used?**    The tag *case outcome detailed (label: CaseOutcomeDetailed)* is a document class with the following categories (designed as drop-down menu). Note that some of the tags may be overlapping. To solve this, priorities are given where necessary:

**[DAMAGES_AWARDED]**  Damages awarded. This includes the award of equitable compensation.

**[DAMAGES_NOT_AWARDED]**  Damages not awarded. This includes the rejection to award equitable compensation.

**[EQUITABLE_REMEDY_AWARDED]**  Equitable remedy(s) awarded. Note to annotators: equitable remedies include specific performance, restitution, recission, rectification, estoppel, account of profits. There is a separate tag for declaration and injunction.

**[EQUITABLE_REMEDY_NOT_AWARDED]**  Equitable remedy(s) not awarded

**[FAMILY_COURT_REMEDY_GRANTED]**  Family Court remedy granted

**[FAMILY_COURT_REMEDY_NOT_GRANTED]**  Family Court remedy not granted

**[SUMMARY_JUDGEMENT_GRANTED]**  Summary judgement granted

**[SUMMARY_JUDGEMENT_NOT_GRANTED]**  Summary judgement not granted

**[INJUNCTION_AWARDED]**  Injunction awarded

**[INJUNCTION_NOT_AWARDED]**  Injunction not awarded

**[DECLARATION_MADE]**  Declaration made (includes both doctrinal and functional declarations). The declaration tag is understood in both a doctrinal and a functional way. It takes priority over BREACH_OF_STATUTORY_OBLIGATION. For example, it applies when a court confirms the validity of an appointment or the breach of an IP right (while the damages flowing from this would be annotated as DAMAGES_AWARDED). It also applies where the court decides on a certain interpretation of the law or a contract. It also applies where court approval is needed to achieve a legal consequence (e.g., that the requirements for convening a meeting are present).

**[DECLARATION_NOT_MADE]**  Declaration not made (includes both doctrinal and functional declarations). Note to annotators: for example, a breach of obligation under the Human Rights Act 1998.

**[STAY_ORDERED]**  Stay ordered

**[STAY_NOT_ORDERED]**  Stay not ordered

**[STRUCK_OUT]**  Application struck out

**[NOT_STRUCK_OUT]**  Application not struck out

**[BREACH_OF_STATUTORY_OBLIGATION]**  Breach of statutory obligation

**[NO_BREACH_OF_STATUTORY_OBLIGATION]**  No breach of statutory obligation

**[APPEAL_DISMISSED]**  Appeal dismissed

**[APPEAL_UPHELD]**  Appeal upheld

**[GUILTY_VERDICT]**  Guilty verdict. This includes contempt of court.

**[NOT_GUILTY_VERDICT]**  Not-guilty verdict

**[CONVICTION_QUASHED]**  Conviction quashed

**[CONVICTION_NOT_QUASHED]**  Conviction not quashed

**[ORIG_DECISION_UPHELD_UNDER_JUDICIAL_REVIEW_REMEDY]**  Original decision upheld under judicial review remedy(s)

**[ORIG_DECISION_QUASHED_JUDICIAL_REVIEW_REMEDY]**  Original decision quashed and/or judicial review remedy(s). Note to annotators: where a decision is quashed on judicial review, remedies may include quashing order, prohibiting order, mandatory order, declaration, injunction, damages.

## E  Topic Model Top Words

Below we list the top words found in each topic given by the topic model.

**Immigration (Topic 32):** leave, application, rules, immigration, remain, decision, paragraph, appeal, secretary, state, united, kingdom, requirements, made, granted, refusal, uk, enter, appendix, policy.

**Financial Services (Topic 35):** investment, transaction, financial, market, bank, client, risk, rbs, transactions, investors, million, fsa, business, trading, clients, investments, barclays, banks, management, fund.

**Employment (Topic 55):** employment, work, contract, employer, employee, employees, employed, working, pay, time, hours, worker, transfer, job, terms, service, business, paid, worked, workers.

**Contract (Topic 95):** agreement, contract, clause, parties, terms, party, agreed, term, obligation, contractual, breach, construction, obligations, effect, agreements, words, rights, contracts, termination, provision.

## F  Evaluation of GPT Models

Below are instructions regarding how we tested the two GPT-based models. GPT-3.5-turbo has a 4k token limit, while GPT-4 has an 8k token limit. We chose to input the same text to all GPT models for a fair comparison, not utilising the larger token limit of GPT-4. Since very few cases fit into the above token limit, we chose to simply segment the cases without overlap into approximately 3k token segments, which were processed independently via the OpenAI chat completion API. Chunks were created such that sentences were added one sentence at a time, until the total size of the chunk exceeded 3k tokens, to ensure we had enough tokens left for the prompt and the response. Tokens were calculated using the TIKTOKEN python library.

The prompts used for case outcome extraction are as follows:

```
{"role": "system", "content": "Please be a legal research assistant to
↪  assist legal professionals in analyzing and understanding legal cases"}
{"role": "user", "content": """Please carefully read the provided text and
↪  identify the sentence(s) that describe the outcome(s) of the case. Copy
↪  and paste the exact quotes from the text. If there are multiple
↪  outcomes, separate the quotes with "[SEP]". Provide the quotes in
↪  chronological order, being concise and avoiding unnecessary information.
↪  If the case outcome is not provided in the text, reply with 'NA'.
Case outcome definition: The result of the case expressed as winning or
↪  losing from the claimants perspective.
Examples of case outcome:
For the reasons set out in this Ruling , we have decided to grant
↪  permission on all of the grounds proposed.
I would dismiss the appeal.
For these reasons the appeal is denied."""
+ legal_case}
```

where LEGAL_CASE is the chunk of case text described above.

We set the temperature to zero, since the goal is to quote text. Both the system message and the prompt were created with the help of GPT, asking ChatGPT to improve an initial prompt which

was created based on our annotation guidelines. This prompt was then improved upon iteratively, by stating to ChatGPT what error the previous prompt caused and asking for an improved version, until satisfactory performance was reached on validation data. For the instances where the annotators annotated multiple parts of text within a chunk as case outcome, those parts were concatenated with the text *[SEP]* separating the outcomes, in order to account for multiple outcomes. This was done both to avoid performing extra parsing of output from the GPT models, and because it makes it straightforward to calculate the word error rate for an entire case, and not just each individual sentence.

# Cambridge Law Corpus: Datasheet

**Andreas Östling**[1]  **Holli Sargeant**[2]  **Huiyuan Xie**[2]  **Ludwig Bull**[3]
**Alexander Terenin**[2]  **Leif Jonsson**[4]  **Måns Magnusson**[1]  **Felix Steffek**[2]
[1]Uppsala University  [2]University of Cambridge  [3]CourtCorrect  [4]Sudden Impact AB

## Motivation

**For what purpose was the dataset created?** Was there a specific task in mind? Was there a specific gap that needed to be filled? Please provide a description.

The Cambridge Law Corpus (CLC) was created to enhance and facilitate research in law, natural language processing and machine learning. The corpus consists of 258,146 cases from UK courts, spanning from 1595 to 2020.

**Who created this dataset (e.g., which team, research group) and on behalf of which entity (e.g., company, institution, organization)?**

CourtCorrect Ltd represented by one of the authors, Ludwig Bull, as director has provided the original dataset (containing the raw data) to the research project. The CLC was then created by the authors as part of a research project based at the Faculty of Law of the University of Cambridge. Associated copyrights, such as Crown copyright, are acknowledged.

**Who funded the creation of the dataset?** If there is an associated grant, please provide the name of the grantor and the grant name and number.

The dataset was created within the UKRI-JST funded research project *Legal Systems and Artificial Intelligence* at the University of Cambridge. The grant reference is ES/T006315/1.

## Composition

**What do the instances that comprise the dataset represent (e.g., documents, photos, people, countries)?** Are there multiple types of instances (e.g., movies, users, and ratings; people and interactions between them; nodes and edges)? Please provide a description.

The corpus consists of a collection of 258,146 court cases from the United Kingdom. The data contains information from the judgements, including the court, judges, parties and legal representation in the case. The main information is the decision of the judge(s). The corpus contains a wide variety of cases from different jurisdictions within the UK, different courts across different levels of court hierarchy (for example, county courts, High Court, Court of Appeal and Supreme Court) and time periods.

In addition to the original court cases, the data also contains annotations of case outcomes for 638 cases annotated by legal experts at the University of Cambridge. This was added since, in the UK, there is no strict formal way in which court decisions are written, and it is therefore, non-trivial to identify the case outcome within the text.

**How many instances are there in total (of each type, if appropriate)?**

There are 258,146 cases in total. We release 638 annotated cases, split into 150 test cases, 100 validation cases and 388 cases for training.

**Does the dataset contain all possible instances or is it a sample (not necessarily random) of instances from a larger set?** If the dataset is a sample, then what is the larger set? Is the sample representative of the larger set (e.g., geographic coverage)? If so, please describe how this representativeness was validated/verified. If it is not representative of the larger set, please describe why not (e.g., to cover a more diverse range of instances, because instances were withheld or unavailable).

The Cambridge Law Corpus (CLC) is a legal corpus with 258,146 court cases from the combined jurisdictions of the United Kingdom. The United Kingdom does not have one single unified legal system. Instead, it has what are called three dis-

tinct jurisdictions, which are England and Wales, Scotland and Northern Ireland. This rule has some exceptions: certain courts or tribunals hear cases from across these three jurisdictions. At present, we provide a sample corpus including 258,146 judgements from the combined United Kingdom jurisdiction and England and Wales.

We chose to exclude certain courts from the dataset, including courts from Scotland and Northern Ireland, and some European judgements published in the United Kingdom. The choice to exclude these cases is for two primary reasons. First, these legal jurisdictions are distinct and this corpus seeks to represent the laws of England and Wales and the combined courts of the United Kingdom. In particular, given the United Kingdom's exit from the EU, we have excluded European judgements. Second, according to the doctrine of precedent, a court is bound by the decisions of a court above it, so we have included the superior courts of the United Kingdom as their judgements are very relevant for the courts in England and Wales.

In terms of completeness and representativeness, we make available all court decisions that are legally and ethically available to us. As courts in the UK do not report all decisions they make, it is currently not possible to know how many other court decisions there are that are not in our dataset. Against this background, this dataset cannot claim to be comprehensive or representative. However, this dataset is the largest dataset currently available on UK court cases and, therefore, making it available can be seen as a major step for the research community.

**What data does each instance consist of? "Raw" data (e.g., unprocessed text or images) or features?** In either case, please provide a description.

The dataset consists of legal cases with the raw text from the case as well as metadata on most cases. Certain cases are OCRed from PDF and/or converted from Microsoft Word format. Metadata includes information on the name of the court, jurisdiction, names of judges and date of the hearing.

**Is there a label or target associated with each instance?** If so, please provide a description.

In addition to metadata described above, 638 cases have annotations at the token level for the outcome of the specific case. Some cases have more than one instance of case outcome. Legal experts have annotated information within the case text for meta-data and juridical information.

In particular, the annotations focused on annotating the case outcome at a general and detailed level. See the main publication's appendix for the annotation guidelines.

**Is any information missing from individual instances?** If so, please provide a description, explaining why this information is missing (e.g., because it was unavailable). This does not include intentionally removed information, but might include, e.g., redacted text.

Case body text, year and court are available for all court cases with some exceptions: (i) the courts might anonymise certain cases due to their sensitive nature, for example those containing graphic content or cases related to child abuse, (ii) cases not available in raw text were OCRed from PDF versions, (iii) non-UTF8 characters were removed to comply with XML format.

**Are relationships between individual instances made explicit (e.g., users' movie ratings, social network links)?** If so, please describe how these relationships are made explicit.

Currently, there are no links between different cases. This also applies to the annotated cases. However, the content of many cases will refer to other cases using legal references.

**Are there recommended data splits (e.g., training, development/validation, testing)?** If so, please provide a description of these splits, explaining the rationale behind them.

The annotated data contains a pre-specified test set of size 150, a validation set of size 100 and a training set of 388, all of which were selected uniformly at random.

**Are there any errors, sources of noise, or redundancies in the dataset?** If so, please provide a description.

Some part of the material has been OCRed by us from original PDF files using Tesseract (v4.1.1). Hence, for some of the cases, there might be OCR errors. There might also be errors in the facts and decisions reported by the courts, as well as possible errors arising from transcription and digitalisation of printed material, particularly in older cases. However, in terms of the dataset, we do not technically view all such instances as errors, since our aim is to make court decisions available as they have been made.

**Is the dataset self-contained, or does it link to or otherwise rely on external**

**resources (e.g., websites, tweets, other datasets)?** If it links to or relies on external resources, a) are there guarantees that they will exist, and remain constant, over time; b) are there official archival versions of the complete dataset (i.e., including the external resources as they existed at the time the dataset was created); c) are there any restrictions (e.g., licenses, fees) associated with any of the external resources that might apply to a future user? Please provide descriptions of all external resources and any restrictions associated with them, as well as links or other access points, as appropriate.

The corpus is self-contained.

**Does the dataset contain data that might be considered confidential (e.g., data that is protected by legal privilege or by doctor-patient confidentiality, data that includes the content of individuals non-public communications)?** If so, please provide a description.

No. The dataset contains court judgments that are already publicly available and free of charge published by the UK Courts, UK Government and various commercial and charitable data providers. Therefore, the dataset only contains data that has already been made available to the public elsewhere.

**Does the dataset contain data that, if viewed directly, might be offensive, insulting, threatening, or might otherwise cause anxiety?** If so, please describe why.

The dataset contains all types of court judgments. There may be judgments that may be difficult for some users to read (e.g., court decisions describing criminal offences). However, the courts are sensitive in their description of challenging cases, and no information should be offensive, insulting or threatening to the usual reader. We cannot, of course, predict the reaction of every reader as the sensitivity of individuals differs. Also, since there are very old cases, these might contain language or words used that might be considered offensive today.

**Does the dataset identify any subpopulations (e.g., by age, gender)?** If so, please describe how these subpopulations are identified and provide a description of their respective distributions within the dataset.

Subpopulations (e.g., grouped by age or gender) are not identified. However, individuals are iden-

tified in certain cases: please see the answer to the next question.

**Is it possible to identify individuals (i.e., one or more natural persons), either directly or indirectly (i.e., in combination with other data) from the dataset?** If so, please describe how.

Yes, individuals' names are included. The names of parties, lawyers, judges and other individuals may be directly identified. In the UK, court judgements are not anonymised and all information is currently publicly available. Parties should expect to be named in judgements because courts uphold the principle of open justice, promote the rule of law and ensure public confidence in the legal system. However, the court will anonymise a party if the non-disclosure is necessary to secure the proper administration of justice and to protect the interests of that party. For example, asylum seekers, victims of violent offences and children's names are, in most circumstances, anonymised. In addition, courts will not make cases available for publication if important aspects such as safety or overriding interests stand in the way of publication—as a result, such cases will not be contained in our dataset.

**Does the dataset contain data that might be considered sensitive in any way (e.g., data that reveals racial or ethnic origins, sexual orientations, religious beliefs, political opinions or union memberships, or locations; financial or health data; biometric or genetic data; forms of government identification, such as social security numbers; criminal history)?** If so, please provide a description.

Yes, court judgements will directly include sensitive information where it is relevant to the proceedings. In some cases, it will be indirectly identifiable. All such sensitive information is already publicly available in the published court judgements, which means that this dataset does not add new information. Given the impossible task of removing all personal or sensitive information from this dataset, as it would reduce the integrity of the research corpus, we follow GDPR exemptions on the processing of sensitive categories. We have implemented robust safeguards and considerations relevant to processing data of this sensitive nature. Details on these safeguards are described in Section 3 of the paper, which focuses on legal and ethical considerations.

**Any other comments?**

N/A.

---



**Collection Process**



**How was the data associated with each instance acquired?** Was the data directly observable (e.g., raw text, movie ratings), reported by subjects (e.g., survey responses), or indirectly inferred/derived from other data (e.g., part-of-speech tags, model-based guesses for age or language)? If data was reported by subjects or indirectly inferred/derived from other data, was the data validated/verified? If so, please describe how.

The data was contributed to the research project by CourtCorrect Ltd represented by Ludwig Bull, one of the co-authors. Felix Steffek, a further co-author, has received the data under a licence agreement with CourtCorrect Ltd. Under the license agreement, CourtCorrect Ltd has no rights to interfere with the dataset making it robustly available to the research project.

**What mechanisms or procedures were used to collect the data (e.g., hardware apparatuses or sensors, manual human curation, software programs, software APIs)?** How were these mechanisms or procedures validated?

The corpus was given to the research project in various formats including plain text, Microsoft Word files and PDF files. All files were then converted to a simple standardised XML-format. PDF files were OCRed using Tesseract version (v4.1.1).

**If the dataset is a sample from a larger set, what was the sampling strategy (e.g., deterministic, probabilistic with specific sampling probabilities)?**

We have not intentionally sampled from a larger set. As already described above, the starting point was a raw dataset containing cases from various courts. Courts do not systematically report all of their cases in electronic form, which means that not all cases from every court are published online. As a consequence, this dataset does not necessarily contain all judgements made by the relevant courts. We further excluded all courts that were not within our target jurisdictions of England, Wales and the United Kingdom. Taken together, the dataset is therefore unlikely to contain every case from all the selected courts. However, we have not intentionally omitted any specific case from the dataset.

**Who was involved in the data collection process (e.g., students, crowdworkers, contractors) and how were they compensated (e.g., how much were crowdworkers paid)?**

The data was given to the co-authors of this research project by Felix Steffek, one of the co-authors, having received it from CourtCorrect Ltd. The material was then processed by the authors and stored in XML format. For the annotation of case outcomes, legal experts at the University of Cambridge were hired and paid through the University of Cambridge Temporary Employment Service according to standard compensations for research assistants.

**Over what timeframe was the data collected?** Does this timeframe match the creation timeframe of the data associated with the instances (e.g., recent crawl of old news articles)? If not, please describe the timeframe in which the data associated with the instances was created.

The court cases included in the corpus are cases from 1595 to 2020 and were obtained by the research project in 2021.

**Were any ethical review processes conducted (e.g., by an institutional review board)?** If so, please provide a description of these review processes, including the outcomes, as well as a link or other access point to any supporting documentation.

Two different ethical reviews of this work have been conducted by two different institutions.

The first of these was conducted at the University of Cambridge. Since the research project is allocated to the Centre of Business Research at the University of Cambridge, the ethical review process of the Centre of Business Research was applicable. The review process consisted of supplying the Research Ethics Committee with the relevant information about this research based on a detailed form to be completed. The information submitted included the involved researchers, the purpose of the research, the research methods, the impact on those affected by the research, the experience of the research team, indemnity insurance, how and for how long data will be stored and the data protection rules to be followed. The Research Ethics Committee discussed these arrangements in a meeting, including the GDPR and other legal and regulatory requirements relating to this project. The discussions of the Research Ethic Committee are confidential and the authors of this paper were not part of the discussions of the Committee.

The Research Ethics Committee decided on the ethical review on 4 April 2022, granting ethics approval. The Committee emphasised that it was satisfied that the relevant requirements were fully met.

In addition, a separate ethical review has been conducted in Sweden by the Swedish Ethical Review Authority. This was done by the Swedish Ethical Review Authority according to Act (2003:460) of Sweden concerning the ethical review of research involving humans. The Swedish Ethical Review Authority is a government agency independent of Uppsala University and assesses whether the research application is in line with Swedish statutes. For this review, we submitted an ethical application containing a general research plan, a CV for the responsible researcher and a standardised form with questions on, for example, research questions and aims, methods used, ethical considerations and a time plan. These documents were then considered by the Swedish Ethical Review Authority.

The ethical review was assigned the number 2022-02063-01 and was approved in May 2022.

**Did you collect the data from the individuals in question directly, or obtain it via third parties or other sources (e.g., websites)?**

We obtained the data from CourtCorrect Ltd, represented by Ludwig Bull, one of the co-authors. Under a licence agreement, CourtCorrect Ltd transferred the data to Felix Steffek, a further co-author, who made the data available to the research project. The licence agreement allows to use and republish the dataset. CourtCorrect Ltd has confirmed that:

1. All cases in the dataset are already accessible for the public.

2. The dataset contains cases from at least three public sources.

3. The dataset has been legally obtained.

4. CourtCorrect Ltd has no power to restrict the use of the dataset by the research project in any way now or in the future.

Our work creates a new corpus by converting the content obtained from CourtCorrect Ltd into a standardised form, adding tags and metadata, adjusting the content to fix errors and remove parts that are excluded, and providing code that enables the corpus' content to be accessed via an API.

**Were the individuals in question notified about the data collection?** If so, please describe (or show with screenshots or other information) how notice was provided, and provide a link or other access point to, or otherwise reproduce, the exact language of the notification itself.

No. It would be very difficult to do this in an ethically sound way. We are legally exempt from notification requirements. See details above, which are given in the answer on personally identifiable information, as well as in the main paper.

**Did the individuals in question consent to the collection and use of their data?** If so, please describe (or show with screenshots or other information) how consent was requested and provided, and provide a link or other access point to, or otherwise reproduce, the exact language to which the individuals consented.

Not relevant: see above.

**If consent was obtained, were the consenting individuals provided with a mechanism to revoke their consent in the future or for certain uses?** If so, please provide a description, as well as a link or other access point to the mechanism (if appropriate).

Not relevant: see above.

**Has an analysis of the potential impact of the dataset and its use on data subjects (e.g., a data protection impact analysis) been conducted?** If so, please provide a description of this analysis, including the outcomes, as well as a link or other access point to any supporting documentation.

Yes. As part of the ethical clearance and considerations, the impact on data subjects has been considered: see Section 3 in the main paper. In brief, the dataset contains publicly available information that does not raise material risks to individuals' rights, freedoms or interests. In any event, our safeguards also act to protect against any risks in further use of the corpus. We implement several mitigation strategies, such as the distribution limitations discussed in the paper, including that we will comply with any request from any data subject that information be removed from the corpus.

**Any other comments?**

N/A.

## Preprocessing/cleaning/labeling

**Was any preprocessing/cleaning/labeling of the data done (e.g., discretization or bucketing, tokenization, part-of-speech tagging, SIFT feature extraction, removal of instances, processing of missing values)?** If so, please provide a description. If not, you may skip the remainder of the questions in this section.

Some standard cleaning, such as removing incorrect non-UTF8 characters, correcting obvious OCR errors and identification of incorrect data has been conducted. We have also split the case into header and body text.

**Was the "raw" data saved in addition to the preprocessed/cleaned/labeled data (e.g., to support unanticipated future uses)?** If so, please provide a link or other access point to the "raw" data.

The original raw dataset has been stored temporarily but will not be made public because it includes content from courts that are not within our scope and content that is proprietary. Since the case references have been included in the corpus, it is possible to obtain the original cases if needed. We want to keep the corpus representation in the XML format as close as possible to the original documents. Hence, we minimise preprocessing of the material and only correct obvious errors in the digital representation to better represent the originals, for instance obvious OCR errors.

**Is the software used to preprocess/-clean/label the instances available?** If so, please provide a link or other access point.

Unfortunately, we are unable to make this available because parts of it are proprietary.

**Any other comments?**

N/A.

## Uses

**Has the dataset been used for any tasks already?** If so, please provide a description.

Two small experiments have been conducted to demonstrate example use cases. First, a Latent Dirichlet Allocation topic model analysis has been done on the corpus. In addition, baselines for case outcome extraction and prediction tasks have been conducted.

**Is there a repository that links to any or all papers or systems that use the dataset?** If so, please provide a link or other access point.

Yes. The repository for the corpus can be found here: HTTPS://WWW.CST.CAM.AC.UK/RESEARCH/SRG/PROJECTS/LAW.

**What (other) tasks could the dataset be used for?**

Some examples might include pre-training language models, fine-tuning, information extraction and summarisation, historical analysis of law, legal applications, etc. Due to its scale, we believe there is a huge potential with this material.

**Is there anything about the composition of the dataset or the way it was collected and preprocessed/cleaned/labeled that might impact future uses?** For example, is there anything that a future user might need to know to avoid uses that could result in unfair treatment of individuals or groups (e.g., stereotyping, quality of service issues) or other undesirable harms (e.g., financial harms, legal risks) If so, please provide a description. Is there anything a future user could do to mitigate these undesirable harms?

It is difficult to anticipate all future uses of the dataset that might be harmful. Unfair uses of this dataset are possible. We require everyone who uses the dataset to use it in legal and ethical ways only. In addition, please see the answer directly below.

**Are there tasks for which the dataset should not be used?** If so, please provide a description.

The dataset can only be used for research purposes and cannot be used for any other purposes such as commercial purposes. Also, we will hold users of the dataset to all relevant legal restrictions including the GDPR. For example, the data cannot be used for automated decision-making processes, including profiling, which produce legal effects or similarly significantly affect a person (GDPR art 22). Research outputs, regardless of their format, must never identify natural persons, legal persons or similar entities.

## Distribution

**Will the dataset be distributed to third parties outside of the entity (e.g., company, institution, organization) on behalf**

**of which the dataset was created?** If so, please provide a description.

Yes. The data will be made public for non-commercial research use.

**How will the dataset will be distributed (e.g., tarball on website, API, GitHub)** Does the dataset have a digital object identifier (DOI)?

The dataset will be made publicly available for research purposes exclusively. Access to the dataset will be granted to applicants who meet the following criteria: (1) the data will be used solely for academic research, (2) users must agree to the terms and conditions, (3) users must provide an ethical clearance form signed by their department or similar, ensuring compliance with ethical procedures when utilising the dataset. These requirements are in place to minimise the potential for misuse of the dataset while maintaining an approach in the spirit of open access. Once these criteria are fulfilled, applicants will be directed to the data download page, where they can access the complete CLC dataset as a zip file. The DOI for the dataset is: 10.17863/CAM.100221.

**When will the dataset be distributed?**

The dataset will be released upon the release of the accompanying paper describing the material together with ethical and legal issues.

**Will the dataset be distributed under a copyright or other intellectual property (IP) license, and/or under applicable terms of use (ToU)?** If so, please describe this license and/or ToU, and provide a link or other access point to, or otherwise reproduce, any relevant licensing terms or ToU, as well as any fees associated with these restrictions.

The dataset will be distributed under an individual licence containing the terms ensuring legal and ethical compliance. The terms can be seen on the repository website HTTPS://WWW.CST. CAM.AC.UK/RESEARCH/SRG/PROJECTS/LAW.

**Have any third parties imposed IP-based or other restrictions on the data associated with the instances?** If so, please describe these restrictions, and provide a link or other access point to, or otherwise reproduce, any relevant licensing terms, as well as any fees associated with these restrictions.

There are no particular IP-based or other restrictions imposed on the data by third parties. Crown copyright is acknowledged as regards the court decisions.

**Do any export controls or other regulatory restrictions apply to the dataset or to individual instances?** If so, please describe these restrictions, and provide a link or other access point to, or otherwise reproduce, any supporting documentation.

There are no export controls or similar regulatory restrictions related to this dataset

**Any other comments?**

N/A.

| Maintenance |
| --- |

**Who will be supporting / hosting / maintaining the dataset?**

The corpus will be continuously maintained by the authors at the University of Cambridge.

**How can the owner/curator/manager of the dataset be contacted (e.g., email address)?**

The manager of the dataset can be contacted at clc@law.cam.ac.uk.

**Is there an erratum?** If so, please provide a link or other access point.

Rather than adopting an erratum for a fixed dataset, we will release multiple versions of the dataset which correct any discovered errors, with updates listed in a changelog.

**Will the dataset be updated (e.g., to correct labeling errors, add new instances, delete instances)?** If so, please describe how often, by whom, and how updates will be communicated to users (e.g., mailing list, GitHub)?

Yes. The corpus will be updated and improved regularly by the research group. We will keep adding more features and cases to the dataset as they become available. We will release new versions of the corpus and will present a changelog on the corpus landing page.

**If the dataset relates to people, are there applicable limits on the retention of the data associated with the instances (e.g., were individuals in question told that their data would be retained for a fixed period of time and then deleted)?** If so,

please describe these limits and explain how they will be enforced.

This dataset does relate to people, with no limitation on availability. Individuals can request to be removed from the corpus by contacting us using information provided on the landing page, and we will comply if they do.

**Will older versions of the dataset continue to be supported/hosted/maintained?** If so, please describe how. If not, please describe how its obsolescence will be communicated to users.

The corpus is versioned and controlled using GIT. Hence, different versions will be stored, although not necessarily publicly. Data will be versioned using semantic versioning where a major version will be changed if the format or structure of data changes will not be back-compatible. Minor versions will be changed if new features are added and patch versions will be changed if errors are corrected in the corpus.

**If others want to extend, augment, build on, or contribute to the dataset, is there a mechanism for them to do so?** If so, please provide a description. Will these contributions be validated/verified? If so, please describe how. If not, why not? Is there a process for communicating/distributing these contributions to other users? If so, please provide a description.

We will allow suggestions and contributions to the dataset. These changes will be quality-controlled by random sampling, and if contributions improve the corpus, they will be merged using GIT for traceability. Information on how to contribute suggestions is available on the corpus's landing page.

**Any other comments?**

N/A.

