# OpenReview forum: "The Cambridge Law Corpus: A Dataset for Legal AI Research"
_NeurIPS.cc/2023/Track/Datasets_and_Benchmarks — NeurIPS 2023 Datasets and Benchmarks Poster_

### Official Review · Reviewer_Hryg · 2023-07-21
**The way we should create and present legal datasets for AI community.**

**Rating:** 9
**Confidence:** 5

**Strengths:**

- The paper describes the CLC dataset, including the number of cases, their period, and the availability of raw text and metadata. Including a datasheet and license information adds to the transparency and reproducibility of the research.

- The authors demonstrate a strong understanding of the UK judicial system, which helps non-UK readers comprehend the dataset. The choice of legal experts as annotators adds credibility to the annotations.

- The authors emphasize the importance of semantic control for future dataset releases, highlighting their commitment to responsible data-sharing practices.

- The paper includes a comprehensive discussion on the ethical implications of using the CLC dataset, addressing privacy concerns and compliance with regulations such as GDPR. The authors have taken measures to mitigate potential harm by allowing individuals to request the removal of specific information.

The annotators have been chosen correctly for the task. `Q1: Did annotators come from the same legal domain or many different legal branches? How diversity the annotated cases were?`

**Additional Feedback:**

It was a pleasure to read the paper and see all necessary supplements added - datasheet, license information, ethical and limitation mentioned.

**Clarity:**

The paper is well-written and effectively communicates the purpose and findings of the research. The authors clearly explain the CLC dataset, the annotations, and the experimental setup. The inclusion of figures and tables further enhances the clarity of the paper.

**Correctness:**

The claims made in the submission appear to be correct. The Cambridge Law Corpus (CLC) dataset is constructed soundly, with over 250,000 court cases from the UK included. The authors provide detailed information on the number of cases, their time span, and raw text and metadata availability. Including annotations on case outcomes for a subset of instances adds value to the dataset.

Regarding the evaluation methods and experiment design, the authors trained and evaluated case outcome extraction models using GPT-3, GPT-4, and RoBERTa models. `Q6: Do you plan to experiment with other LLMs such as llama/llama2, falcon, or LLMs fine-tuned with legal documents?`

It would be helpful to have more information on the prompt creation process, including any iterations or improvements. Additionally, insights into selecting examples/definitions for prompts would enhance understanding of the experimental setup.

**Documentation:**

The paper provides sufficient detail on the data collection and organization of the CLC dataset, including information on the raw text, metadata, and annotations. The authors also give a datasheet and license information, which adds to the transparency and reproducibility of the research. `Q5: I could miss it, but where can I find explicit mention of the availability and maintenance of the dataset in the future? What is the plan for updates and corrections?`

**Ethics:**

It is an excellent practice to confirm legality and ethics with the university entity - `the legality of this project as affirmed by the ethics approval from the University of Cambridge`.

All ethics checkpoints are checked. Authors even mentioned situations such as `Any individual may request the removal of a case or certain information which will be immediately removed` that complies with GDPR and national legislature.

The authors decided that `Access to the corpus will be restricted to researchers based at a university whose head of department confirms that ethical clearance for the research envisaged has been received.` it would be even greater to provide an example of how we can prepare this confirmation.

However, still, data contains personal data, hence it could be potentially used in a harmful way. However, the authors tried to do everything to mitigate these issues. It is worth mentioning `In the UK, court cases are not anonymised.`, however, we must remember that there are nations where court decisions MUST be anonymized, and in these cases, it could mean more restrictive pre-processing to dataset creation. Not only France has this kind of anonymization obligation.

**Limitations:**

`The original documents are stored separately for quality control purposes.` - `Q4: Did you do any manual quality control over OCR or word files extraction of texts? Especially for the OCR part, for older cases could be potentially problematic.`

**Opportunities For Improvement:**

- The paper could provide more details on how the prompts for GPT-X models were created. Knowing if multiple prompts were tried and how the authors improved them besides using ChatGPT would be helpful. Additionally, insights into selecting examples/definitions for prompts would enhance understanding of the experimental setup.
- `Q2: Do you think using ChatGPT to create prompt is better than manual creation and testing?`

Further information on the manual quality control process for OCR and word file extraction, especially for older cases, would be valuable to assess the reliability of the extracted texts. `Q3: Do you think OCR went well on medieval judgments? Can you provide some examples of OCR quality for them?`

It would be great to write a little more (even in the appendix) about the ethical and legal consultation process at the university level. It could influence other researchers to do it more rigorously and more often.

**Relation To Prior Work:**

The authors also highlight the unique challenges of legal data and emphasize the need for responsible data-sharing practices. While more explicit discussion on the relationship to prior work would be beneficial, the paper adequately positions the CLC dataset as a valuable contribution to the legal AI research community.

**Summary And Contributions:**

"The Cambridge Law Corpus: A Corpus for Legal AI Research" introduces the Cambridge Law Corpus (CLC), a comprehensive dataset consisting of over 250,000 court cases from the UK. The authors provide annotations on case outcomes for a subset of cases and use this annotated data to train and evaluate case outcome extraction models. The paper also addresses legal and ethical considerations surrounding using the corpus. Overall, the submission significantly contributes to the legal AI research community by providing a high-quality dataset for legal text analysis.

High-quality datasets for legal AI are still needed, and the authors want to change this, especially if we consider the domain as complicated as legal text.

Legal data involves unique challenges, including complying with regulations such as the GDPR legislation and appropriately addressing privacy and other ethical matters. Hence, creating legal domain datasets is even more critical and complex.

This work introduces a corpus containing, in its present form, 258,146 legal cases from the UK, available for research. We release the corpus and annotations of case outcomes for 638 cases.

---

> ### Author Response · Authors · 2023-08-17
>
> Thank you very much for your review! We are particularly delighted to see our emphasis on ethical and responsible release of this dataset acknowledged - we also feel that this is very important for the community. Below we address some of the questions:
>
> **Opportunities For Improvement:**
>
> **Legal Domain of Annotators.**
> Thank you for this question! Let us provide some additional information while respecting the anonymity policy. All annotators possess a law degree. In addition, some have higher qualifications in law, and are qualified solicitors. The annotators had broad domain knowledge in terms of areas of law. The annotators generally had more civil law experience than criminal law, although that provided a wider area of legal knowledge that aided in the considerable annotations required for civil law judgments (i.e., there are more civil courts and cases in the CLC than criminal).
>
>
> **Details on Prompts, and Creating Prompts via ChatGPT.**
> Thank you for this question! Our prompt creation process is at present described in brief in the appendix. Our final prompts and prompt-creation process involves starting out from an initial prompt written by hand, asking ChatGPT how to improve the prompt, testing out the new prompt with the improvements, and repeating the process recursively. We settled on this process because we found it more effective than manual prompt creation. We recognize that this process is very important, and will add an additional short description of it to the main text. We will also add further details to the appendix.
>
> **Quality Control for OCR.**
> This is a great idea! We performed a very quick preliminary check, by sampling five random paragraphs and manually checking them for errors. At the character level, this resulted in three OCR errors, including punctuation and layout, out of 1285 non-whitespace characters. This corresponds to a rate of roughly 0.2% character-level errors. For the next version of the paper, we will perform a more rigorous estimate by taking a random sample of 40 pages of the PDF documents and manually writing down the text for a random sample of either paragraphs or lines within these pages. This will allow us to produce a more formal estimate of OCR error.
>
> **OCR: older (incl. medieval) judgments.**
> Older judgments like this have been transcribed from the original decisions by the courts, and have been reprinted. Such judgments are usually available in digital form. As a result, OCR for these cases is very similar to that of other cases typewritten on paper. Throughout our corpus, all OCR is performed on printed text - none of it, to our awareness, is performed on handwritten or otherwise non-standard documents.
>
>
> **Ethical and Legal Considerations: University Level Process.**
> Thanks for this question - additional information for the university-level process is provided in the datasheet. In short, at one of our universities, the review process consisted of supplying the relevant information about our research based on a detailed form, including the researchers involved, purpose, methods, impact on those affected by the research, research team experience, indemnity insurance, how and for how long the data will be stored, and the data protection rules that must be followed. The Research Ethics Committee, which is a university body that does not include the authors, met and discussed these arrangements, including the GDPR and other legal and regulatory requirements. In order to make this description easier to find, we will add a sentence to the manuscript which points out that it is described in more detail in the datasheet.
>
> **Limitations**
>
> **OCR: manual quality control.**
> Please see the prior reply regarding OCR quality. Using a very rough preliminary estimate, we believe the OCR error rate to be about 0.2%, and will perform a more formal estimate for the next version of the manuscript.
>
>
> **Correctness**
>
> **Other LLMs.**
> Thanks for the suggestions - we are also interested to know how these models perform in this setting! However, as the primary focus of this submission is to introduce the CLC dataset, we selected a relatively small range of representative models (RoBERTa-based and GPT-based) as initial empirical attempts to showcase the potential use of the CLC for two different settings, in order to avoid excessive costs and focus our effort on dataset quality. Thus, we believe that experimenting with other LLMs is better served as a direction to explore as a next step in future work.
>
> **RESPONSE CONTINUED BELOW**

---

> > ### Author Response · Authors · 2023-08-17
> >
> > **RESPONSE CONTINUED FROM ABOVE**
> >
> > **Documentation**
> >
> > **Maintenance.**
> > Thanks for the question! The details regarding availability and maintenance arrangements of the dataset can be found in our datasheet (see the “Distribution” and “Maintenance” sections). To briefly summarise, the CLC data will be made publicly available for research purposes, under an individual licence containing the terms ensuring legal and ethical compliance. We plan to continuously maintain and further develop the corpus by the research group. We will keep adding more features and new cases to the dataset as they become available and present a change-log on the corpus hosting site. We will add an additional reference to the datasheet to the main manuscript to make this easier to find.
> >
> > **Ethics**
> >
> > **Example of Clearance.**
> > Thanks for this point! Our process is as follows: the user goes on our website, and enters their information by filling out a web form, attaching their ethical clearance PDF given by their university. Please see the form which has been **added to the website**. After this, they receive an email with instructions on how to download the data. Please also note that we have modified and relaxed some of our licence requirements to reflect feedback from the other reviewers, in order to maximise ethical access to the dataset. An example submission form has been added in the form of a link posted on the website.
> >
> > **Anonymity in nations other than France.**
> > Thanks for pointing this out! We agree and are very grateful for your consideration of the use of the dataset abroad. We are covering more restrictive laws in other countries by requiring users not only to apply the laws of the UK but also the laws applying in the jurisdiction that might additionally apply to them. We will add an additional comment to the manuscript noting that restrictive privacy frameworks are not limited to France.

---

> > > ### Comment · Reviewer_Hryg · 2023-08-31
> > > **Final scores**
> > >
> > > Thanks for addressing my comments. I would stick with scores and add my point of view on why my rate is higher than others.
> > >
> > > 1. Licensing - Of course, in the research community, we want to have licenses as open as possible; however, I like the legal process and analysis of why various parts of the dataset (text, pdf, ocred docs, etc.) in specific conditions could not be licensed with for example MIT license. Moreover, in responses to the review, the authors explained why this matter is tricky and not so straightforward.
> > > 2. For me, the dataset part of the paper is more important than any analysis of it. The analysis part of the paper looks for me more like a starting point. The dataset enables more researchers to conduct diverse types of research not only simple topic modeling or classification but using topics to analyze them for concrete directions and use cases.
> > > 3. The paper addresses the problem of how complex cases of datasets should be treated and processed at the universities. This is still not a common effort taken into consideration in dataset creation processes.

---

### Official Review · Reviewer_JUfX · 2023-07-21
**New legal corpus with careful legal consideration**

**Rating:** 4
**Confidence:** 3
**Correctness:** yes
**Clarity:** Reasonably.

**Strengths:**

The authors make new large scale legal corpus that consists of UK precedents with careful legal considerations.

**Additional Feedback:**

There is no.

**Documentation:**

Yes

**Ethics:**

No.

**Limitations:**

.

**Opportunities For Improvement:**

- It will be great if the additional details about the preprocessing step is available. For instance, there is no information about OCR error rate.
- It would be great if additional legal tasks are provided in addition to the case outcome extraction task which seems to be quite straightforward.
- The accuracies of two GPTs could be higher if they generate the label for individual sentences under few-shot setting rather than generate entire target sentences from whole documents. Also, WER metric seems to be difficult to interpret.

**Relation To Prior Work:**

Some previous works are not cited.
For instance, Pile of Law: Learning Responsible Data Filtering from the Law and a 256GB Open-Source Legal Dataset, NeurIPS, 2022

**Summary And Contributions:**

New valuable large scale (250k)  case corpus from UK courts.
Additional  638 annotations for case outcome extraction.

---

> ### Author Response · Authors · 2023-08-16
>
> Thank you for your comments! Below we address the relevant questions:
>
> **Opportunities For Improvement:**
>
> **OCR error.**
> This is a great idea! We performed a very quick preliminary check, by sampling five random paragraphs and manually checking them for errors. At the character level, this resulted in three OCR errors, including punctuation and layout, out of 1285 non-whitespace characters. This corresponds to a rate of roughly 0.2% character-level errors. For the next version of the paper, we will perform a more rigorous estimate by taking a random sample of 40 pages of the PDF documents and manually writing down the text for a random sample of either paragraphs or lines within these pages. This will allow us to produce a more formal estimate of OCR error.
>
>
> **Additional legal tasks.**
> While we agree that defining additional legal tasks would be interesting, our focus was on providing a dataset rather than designing comprehensive benchmarks for machine learning models. We especially focus on handling the legal and ethical aspects required to publish such a dataset legally. Thus, we believe that there is not enough space available in the manuscript to handle additional benchmark tasks, and that this would be better deferred to follow-up work.
> We also want to note that the case outcome extraction task is much less straight-forward than we first anticipated. As an example, due to the highly unbalanced data models like RoBERTa and BERT performs poorly, which makes this task more interesting than might first meet the eye.
>
>
> **GPT model accuracy.**
> Thank you for this idea. While this is a standard approach that one could certainly try, there are two reasons we did not pursue this. (1) In our setting, the case outcome is not necessarily a full sentence, but can instead be part of a sentence. (2) For case outcome extraction, context - including long-range context - is extremely important. Part of the reason for this is that judgments will often refer in-text to other judgments, either by different courts or by different judges in the same case. Let us give some examples:
>
> * It would be common for a superior court to write: “In their judgement, the lower court held: … the Claimant(s) are unsuccessful in their claim.” In such cases, the judgement of the lower court may often be presented *without any quotes*, potentially for paragraphs at a time. The superior court may then decide that it will overturn the decision of the lower court and write: “I would allow this appeal and the Claimant(s) are successful in their claim”. Without context, both sentences may be mistaken for the ‘Case_Outcome’.
>
> * Further, in cases with multiple judges presiding, it is common to see (out-of-context) conflicting statements such as one judge who writes “I would allow the appeal” and another judge who writes “I would refuse the appeal”. Identifying which statement belongs to the majority judgement that is binding and is the true ‘Case_Outcome’. To identify this, one must disentangle it from the dissenting opinion of one judge, which requires discerning long-range context.
>
>
> For this reason, if we want this approach to perform well, we would need to use a very large amount of tokens for GPT to provide enough context, which would make it significantly more expensive than other approaches.
>
> **Relation To Prior Work**
>
> **Additional References**.
> Thank you for the additional references - we very much appreciate this! We have added these to the manuscript.
>
> **Limitations and Other Issues**
>
> We would be very grateful for some additional clarification regarding the limitations and any other remaining issues that you wanted to see addressed, particularly since the text of your review seems to us to be somewhat more *positive* than the score that was given.

---

### Official Review · Reviewer_Sw7Y · 2023-07-24
**UK Court Case Corpus for Legal AI Research**

**Rating:** 7
**Confidence:** 3
**Clarity:** The paper is clear, well written and …

**Strengths:**

This paper has a clear motivation that would be interesting for many in the community. Legal AI research has received increasing attention due to its potential impact in the real world, and one of the major barriers for research in this field is to have more high quality data. A large legal corpus in an easy-to-work-with format with annotations on case outcomes would be an important contribution to the community.

In addition, given the sensitivity of the data, this paper provides great ethical guidelines and addresses the privacy legal concerns well. It also proposes an annotation guideline for UK case outcomes - both could be served as examples for future research.

Overall, the paper is clear and well-written, and based on the sample dataset, the dataset and experiment results are reasonably well documented.

**Additional Feedback:**

Currently I do not have additional feedback that are not covered.in the sections above.

**Correctness:**

To the best of my knowledge, the legal corpus dataset is constructed in an appropriate way. I have also voiced my other concerns in the section above.

**Documentation:**

The authors provided appropriate documentations on the sample dataset, including URL access, licensing and maintenance plan. I would appreciate if the authors could provide more data samples, as the current sample dataset does not contain many cases closer to present (post 2010), or any sample of the metadata.

**Ethics:**

The paper provides a very thoughtful approach towards consent, privacy, GDPR compliance, and potential misuse. The proposed management and distribution method is appropriate given the data sensitivity.

**Limitations:**

In the main paper, the authors discussed the potential error in the corpus creation process, such as OCR errors, annotation errors etc. I believe that this paper could benefit from more discussions on its limitation in the appendix, including the case selection process, how representative it is of the UK legal landscape, if there are potential biases, etc.

**Opportunities For Improvement:**

The paper could benefit from discussions with regards to the following questions:
1. Discussion of the dataset source: the authors mentioned that "the original cases of the Cambridge Law Corpus were supplied by the legal technology company CourtCorrect in raw form", without further explaining the details about CourtCorrect. For example, where did the case data come from at CourtCorrect? What was their method of creating the original case dataset? Did they scrape the public court records on the Internet? Why is the dataset from CourtCorrect trustworthy?
2. How are the cases selected, given that there is a much larger number of cases over the past decades and even centuries at the courts mentioned in the dataset?
3. In the discussion of the topic model analysis, why is the topic model analysis of CLC representative of the legal trends in the UK? There could be selection biases, as it is clear in Figure 2 that there are too few cases in certain years that skew the graph. If the number of cases is much higher in the 1990s onwards, should the topic model analysis be limited to certain time period where there are enough samples?
4. What criteria did the authors use to select cases from these specific time periods? Why did the authors include those old cases? Is it because of their importance as precedents? From the appendix, it is clear that some older data points skewed the distribution of certain metrics (e.g. number of tokens per case per year).
5. Corpus creation process: the authors mentioned that "changes to the corpus are quality controlled through a random sample of the edits made." Could the authors elaborate more on the QA process? How many samples would authors take for each update? Are there better methods than a random spot check?
6. In the case outcome extraction task, the authors used WER as the main performance metric, which is a standard, simple but useful metric. However, it treats all errors with equal weight and sensitive to reference variability. Would it be helpful if WER is used along with other metrics, such as BERTScore?
7. To further demonstrate the potential impact of this dataset, would it be possible to add performance comparison for models pre-trained / finetuned using the dataset vs. standard models?

**Relation To Prior Work:**

To the best of my knowledge, this work discusses how it differs from previous contributions.

**Summary And Contributions:**

This paper introduces a new dataset, the Cambridge Legal Corpus, with over 250,000 court cases in UK for legal AI research. The paper provides the corpus in an easy-to-work-with format and releases the dataset with semantic versioning that helps with future maintenance and improvements. It also goes through a thoughtful privacy and ethics discussion regarding the management and distribution of the corpus given the sensitivity of the data. The paper provides two sample use cases for the dataset: case outcome extraction and topic model analysis, and creates the first benchmark on case outcome extraction.

Overall, the paper is clear, well motivated, and presents a legal corpus dataset that could be very helpful for future research in legal AI. However, some of the discussions regarding the dataset source and creation process are not as clear and can be further improved.

---

> ### Author Response · Authors · 2023-08-16
>
> Thank you very much for your review! We are delighted to see our work seen as an “important contribution to the community” and particularly appreciate your review’s emphasis on the importance of handling legal and ethical aspects appropriately, which we see as central to our work. We address the key questions below:
>
> **Opportunities For Improvement:**
>
>
> **Dataset source.**
> CourtCorrect confirmed in the licence agreement that we entered into that the cases are sourced from public sources where either individual consent has been obtained or where public statements allowed download. We have verified this through having seen consent obtained by CourtCorrect where needed. The authors have worked with CourtCorrect for a number of years and can confirm trustworthiness of the company, and are happy to provide more details, which cannot be stated here due to submission anonymity, to the AC if needed.
>
> **Selection of Cases**
> We did not explicitly select any cases in or out, except by jurisdiction (we exclude Scotland and Northern Ireland - details are given in the manuscript) and are publishing all cases that we were able to obtain. We believe that **availability in digital form** is the most significant, though by no means the only, variable which determines whether or not a given UK case is available in our corpus.
> * Note that not all judgments are consistently published or made otherwise available by the courts, even though access to court judgments is fundamental to the principle of open justice (discussed in Section 3) under the UK common law system. The Supreme Court and Privy Council generally publish their decisions comprehensively. However, lower court decisions may not always be captured by various sources and some discretion is held by judges or the judicial press office.
> * To give an example of the limited digitalization of court cases, a study found that between 2015 and 2020 only 55% of 5,408 administrative court judgments identified by a commercial subscription service appeared on the website of the BAILII (Somers-Joce et al. 2022; Byrom 2023).
>
> We thank the reviewer for asking this question and we will clarify in more detail in the paper. Specifically, we will add an extra subsection to the appendix to describe the typical way through which a court decides whether or not to publish the data, and what goes into these decisions, together with suitable references for readers who need more detail.
>
>
>
> **Topic Model: representativeness**
> This is a good point: there is indeed a selection bias in that the courts decide which cases to make public (see above), in a manner that is not fully transparent and can also have changed over time. On this basis, we instead view the topic model as **representative of the data available in our specific dataset** - an important way of understanding the corpus’ content. We will amend the text to better emphasise this.
>
>
> **Time Periods.**
> We do not select or exclude any data based on time period. The only selection we made was to exclude cases based on jurisdiction (see response in Selection of Cases). Otherwise, the corpus collects every case we have access to. We aim to add further cases to the corpus as they become available. The corpus contains information on date of hearing and hence users can choose what cases they want to include in their research.
>
>
> **QA process and random sampling.**
> The QA process consists of drawing a random sample of edits from changed files. The size of the sample would be selected to balance quality with the time required to manually check changes. Previous experiences indicate that roughly 50-100 edits is usually sufficient to strike a balance between quality and cost. In addition to random spot checks, we apply certain forms unit testing to the data, for instance XML validation, for quality control. We plan to further expand the amount of automated tests as the corpus grows.
>
> **Metrics and Word Error Rate.**
> This is a good idea! We agree that WER is not perfect, but also think that in a corpus like this one should start with the simplest and most fool-proof techniques. However, we also think adding BERTScore is a good idea: We therefore calculated BERTScore for both GPT models. GPT-3.5 got F1: 0.7875, Recall: 0.8252, Precision: 0.7616. GPT-4 got F1: 0.84, Recall: 0.9148, Precision 0.8731. We will also add metrics for the other models once their respective computations finish. We will incorporate all of these additional metrics to the manuscript.
>
>
> **Performance comparison: off-the-shelf vs. pre-trained models**
> This is a great idea - we agree that this would be an interesting next step, but due to the compute and infrastructure required to do it properly, we believe this is better suited for exploration in follow-up work.
>
> **RESPONSE CONTINUED BELOW**

---

> > ### Author Response · Authors · 2023-08-16
> >
> > **RESPONSE CONTINUED FROM ABOVE**
> >
> > **Limitations**
> >
> > **Case Selection Process and Representativeness.**
> > This is an excellent idea: as discussed in the prior response involving selection, we will add an extra subsection to the appendix to describe the typical way through which a court decides whether or not to publish the data, and what goes into these decisions, together with suitable references for readers who need more detail.
> >
> >
> > **Documentation**
> >
> > **More data samples.**
> > Thank you for raising this issue. We have **increased the size of the sample dataset to add five new cases** from the past ten years across a variety of courts. This addition can also provide a clearer overview of how the CLC dataset works. This is now available on the website.

---

### Official Review · Reviewer_Z7DF · 2023-07-25
**Review of Cambridge Law Corpus, an annotated UK court decision dataset**

**Rating:** 5
**Confidence:** 4

**Strengths:**

Recent progress in LLMs prompted in natural language has not yet led to many public datasets and benchmarks providing rigorous evaluation of LLMs' capabilities with respect to judicial decisions. The present work makes progress towards curating UK court decisions and benchmarking LLMs on the task of extracting the case outcome from the full text of the decision. This work appears to be amongst the first annotated datasets of UK court decisions, and constitutes a novel contribution in an important research direction.

**Additional Feedback:**

I have made below a few other comments, mostly as suggestions of possible future work.

* The paper contains no task / benchmark to evaluate the LDA topic models used in Section 4.2. The reference Blei et al., 2003 cited in the submission provides one possible metric -  the perplexity of held-out documents.

* In the US, so-called "briefing" is used to summarize court decisions. This process identifies other key information, in addition to the final outcome, relevant to the precedential value of the case, such as the key facts, issues, and the holding and its rationale for each issue. I expect that there is a similar convention for summarizing judgments in the UK. It would be useful to have this information annotated as well.

* Would in-context learning (providing several examples of identified judgments in the prompt) improve the accuracy of GPT?

* In addition to predicting outcome labels, there is also an opportunity to evaluate generative models on the outcome prediction task by masking the outcomes that have already been annotated and asking the model to generate the predicted outcome.

**Clarity:**

The two-step RoBERTa pipeline and the chunking of the decisions to satisfy the maximum token limit of the GPT models appear to be crucial steps in terms of processing these long court decision. I think these aspects of the benchmark can be described a little more extensively; see also items f. and g. above and references therein for further context and relevance of multimodal models. As noted in item d. above, I suggest developing a single evaluation metric for comparing the RoBERTA and GPT models on the same task.  Apart from the suggestions made elsewhere in this review, I think the paper is relatively clear.

**Correctness:**

The results in the paper seem sounds, but as noted, I suggest addressing items 1. and 2. below to ensure reproducibility of the results and accessibility of the dataset. Also, as noted in item e. above I couldn't reconcile the WER>3 score, which suggests high error rate, with the claim in Section 4.1 that GPT exhibits strong performance.

**Documentation:**

1. Reproducibility: According to the license field for this submission on OpenReview, the code is subject to the MIT license. However codebase does not appear to be available. I reviewed the supplementary materials and via the project website  https://www.cl.cam.ac.uk/research/srg/projects/law/.

2. Access/license/long-term preservation/maintenance: A sample of the dataset is shared via https://www.cl.cam.ac.uk/research/srg/projects/law/ Access to the full dataset requires an application and is subject to what appears to be a restrictive license; I listed the key restrictions below.
i. Access only by application by assistant professors (or higher rank academics) who are employed full-time by an officially recognised university;
ii. The application must include a research plan and list all members of the research group who need access;
iii. The applicant must submit an electronic copy of the ethical approval for their research plan obtained from their university (or other authority);
iv. The applicant agrees to immediately remove any data from the CLC as requested by one of the authors. This may involve deleting the entire CLC; and
v. The Dean of the Faculty (or equivalent authority) of the researcher confirms all of the above requirements and agrees that the Faculty (or equivalent) will **indemnify** all the authors of the CLC paper for any costs incurred (including litigation and alternative dispute resolution) as a result of the applicant’s and related team members’ activities. As part of this, the Dean (or equivalent authority) needs to provide their institutional contact details.

It not clear why these requirements must be imposed given the fact that the UK court decisions are licensed under the open government license and are publicly available. Also Section 3 of the submission says that the personal data in them is exempt from the requirements of the UK General Data Protection Regulation.

Under the current restrictions independent researchers and students/non-tenure track researchers without a supervising professor can't access the dataset. Also, it seem unlikely that other universities will give an unlimited indemnity (essentially writing a blank check) to the authors, which will likely prevent many professors from accessing the dataset. Finally, the authors' right to require the users to delete the entire dataset is inconsistent with the goals of long-term preservation and version control.

The submission mentions that the original court decision were supplied by a 3rd part legal technology company. My main question relating to this the following:
*Are the above terms and conditions motivated at least in part by the fact that the cases were obtained via this company and not directly from the UK government?
*Would the authors be able to make publicly available under a standard license, like Creative Commons, at least a portion of the dataset that is made public and can be obtained directly from the UK government? For example, the paper mentioned said that the UK National Archives made post-2003 cases available.
*Can older cases involving corporate litigants only (which are less likely to implicate the rights of natural persons) be released publicly under a standard license as well?

3. The required author statement that they bear all responsibility in case of violation of rights, etc appears to be missing.

4. A persistent identifier: As the dataset is not public, no DOI or other permanent identifier is available.

**Limitations:**

Further to item a. above and items 1. and 2. below, I suggest revisiting the access / license restrictions and considering the possibility of releasing at least part of the dataset freely to the public under a standard nonrestrictive license. Also, I suggest open sourcing the training/inference/evaluation codebase to ensure reproducibility.

**Opportunities For Improvement:**

In my view, there are several important opportunities for improvement.

a. The principal issues are the restrictive license terms/access restrictions and the apparent lack of open source codebase; see items 1. and 2. below. Would it be possible to release freely to the public under a standard license, like Creative Commons, at least a part of the dataset, e.g., post-2003 cases publicly available from the UK National Archives and possibly older cases between legal entities/corporate disputes?

b. Except as described in the next paragraph, the paper does not evaluate the models on the case outcome **prediction** task, which has been extensively studied in the literature (see, e.g, Chalkidis et al 2019 referenced in the submission and a recent survey in Feng et al., Legal Judgment Prediction: A Survey of the State of the Art, IJCAI-22 Survey Track.) Since the present dataset actually includes detailed case outcome labels, those labels appear to be unused and are not part of the main benchmark/task.

c. Notwithstanding the previous paragraph, Appendix C says that using train/val/test splits belonging to specific courts in an ablation study leads to lower accuracy of case outcome **predictions** of the BERT and T5 models relative to the accuracy of those models with splits based on random shuffles of the decisions. It is unclear why the ablation study was performed using the case outcome prediction task, which currently is not benchmarked in the main paper. (The case outcome **extraction** is instead the focus of the submission.)

d. The paper does not develop a common evaluation metric for comparing the pre-trained RoBERTA models (accuracy and F1 scores) with the GPT models (word error rate score) on the same case outcome extraction task.  (Computing accuracy = 1 - WER does not seem meaningful since WER>3 in this paper.)

e. The discussion of Table 2 in Section 4.1 describes strong performance of the GPT models, but this claim does not appear to be corroborated by the WER scores >3.

f. The dataset contains XMLs with the plain text of the court decisions even though the original decisions have been provided in MS Word and PDF. The formatting information in MS Word and visual cues in PDFs would allow multimodal models, such as GPT4, to better process long multimodal input/create hierarchical representation of long text. Can the original MS Word and PDFs be included in the dataset to preserve the multimodal nature of the data? In what format do the UK National Archives release the post-2003 decisions?

g. Further to the previous paragraph, the present paper effectively creates a hierarchical representation of long decision by using a two-step RoBERTA pipeline and a GPT-based chunking procedure described in Appendix F. Using formatting in MS Word and visual clues in PDF may allow multimodal models like GPT4 to achieve a more contextually coherent chunking than the split into 3k token chunks described in Appendix F.  See, e.g.,
* Hegel et al., The law of large documents: Understanding the structure of legal contracts using visual cues. In Document Intelligence Workshop at KDD, 2021 and
* Rao et al., MarkupMnA: Markup-Based Segmentation of M&A Agreements.


**Relation To Prior Work:**

A more robust literature overview relating to legal datasets, and particularly datasets with annotated court decisions, should provide some guidance, including best practices on licensing/access terms and public codebase release. See, e.g,
* LexGLUE dataset containing annotated certain US and EU court decisions, and
* Korean court judgements in Hwang et al, A Multi-Task Benchmark for Korean Legal Language Understanding and Judgement Prediction, NeurIPS 2022 Track Datasets and Benchmarks.

Also as noted in items f. and g. above, the pipeline based-RoBETRa and chunking of GPT inputs can be contextualized in terms of other literature addressing maximum token limits in the context of long legal documents (as well as in terms of potential for using multimodal models to process long multimodal documents.)

While this paper appears to be amongst the first annotated dataset of UK court decisions, I suggest that the authors explicitly clarify/confirm in the literature review if there exist any other UK court case datasets.

**Summary And Contributions:**

The submission describes the UK court system and introduces a dataset containing over 250,000 UK court judgments (decisions) along with expert annotations of the outcomes in 638 decisions. The cases go back as far as the 16th century, but most of them are from the 20th and 21st century (courts from Scotland and Northern Ireland are excluded).  The text of the decision with the outcome of the case is marked in a separate file along with labels classifying the outcome. The paper benchmarks two popular language models (RoBERTA and GPT) on the task of extracting the outcome of the case from the full text of the decision.  The dataset has bespoke access and license terms.

---

> ### Author Response · Authors · 2023-08-18
>
> Thank you for your detailed comments! We are thrilled that you thought that our work “constitutes a novel contribution in an important research direction” - thank you for these kind words! Below we address some of your questions:
>
> **Opportunities For Improvement**
>
> **Licence.**
> Thank you very much for your comments, on the basis of which **we have made some amendments regarding the licence for the dataset** (regarding the code’s licence, please see below). Let us first summarise the questions:
> * Is it possible to release the data in full, or in part, under a more permissive licence?
> * If not, can older company law cases, or those that are available from the UK National Archives, be released under a more permissive licence?
> * How does our licence compare to those used by other dataset releases in other jurisdictions?
>
> These questions are *critically* important to our work, let us address them in more detail below. First, though, let us comment on how we reached the licensing requirements. This was done by balancing two competing factors: (1) making the corpus as widely available as possible while (2) at the same time respecting the legal and ethical limitations set out in the paper and the data questionnaire. The key reason for these limitations is the following combination of factors, namely that (a) personal information is contained in the dataset, since **UK judgements are not anonymised**, (b) the UK in particular has **strong privacy protections such as the Data Protection Act / UK GDPR**, and (c) given the legal and ethical background of our authorship group, making our dataset **released in a legal and ethical manner is a key area of focus**. Let us provide some details:
> * The UK government, via the National Archives, currently only makes court decisions available in a very restricted way. (1) Users are restricted to 10 cases per transaction and 100 cases per month, (2) bulk downloads are prohibited and (3) usage is restricted to non-commercial private use and educational purposes. The UK government does not explain the reasons for their limitations. However, it is fair to assume that they aim to prevent illegal and unethical use of court data, e.g., the profiling of persons prohibited in Art. 22 of the UK GDPR and other breaches of data protection law.
> * The main reason for the restrictions we impose is that - contrary to the UK government - we are making a bulk court cases dataset available. This bulk nature of the dataset triggers restrictions under data protection law such as the UK GDPR as the bulk dataset could be used for prohibited activities such as personal profiling. When we mention in Section 3 of the paper that our dataset benefits from exemptions, this means that **legal exemptions apply as a result of the licence restrictions we impose**. If we would not impose the licence restrictions we suggest, the access to the dataset would not benefit from legal exemptions such as those offered by the UK GDPR for research-only purposes. Put differently, **if we were not to impose the restrictions we propose, there would be a considerable risk of our dataset being illegal and unethical.** This explains why we cannot make those cases in the National Archives available under a more permissive licence.
> * Against this background, the full corpus is currently restricted to be used for research, and ethical assurances such as an ethics clearance have to be submitted for access. **This mirrors similar restrictions used in other works, for instance medical datasets and those with personal information**, that have previously been published within the NeurIPS dataset track.
> * Even with restrictions, note that compared to the UK government, we make our dataset more widely available while at the same time taking proportionate measures to avoid illegal and unethical use. We have also decided to adapt our licence to better capture the tradeoffs raised by referees, which we will describe below.
>
> **Older corporate cases.** With the above in mind, given it is not possible to release the corpus without some proportionate restrictions, let us address the remaining question regarding older corporate cases. For these, note that the first company laws were only introduced in the UK in the latter part of the 19th century, most company-related cases have been decided by the courts in the 20th and 21st centuries. As a result, there are not many old company law cases. Moreover, company law cases concern natural persons such as directors, shareholders, employees and company secretaries, which are directly protected by the data protection laws and ethical requirements we need to respect. This makes such a potential release concerning older corporate cases fall under the same consideration which lead us to our licence.
>
> **RESPONSE (INCL. LICENSE UPDATES) CONTINUED BELOW**

---

> > ### Author Response · Authors · 2023-08-18
> >
> > **RESPONSE CONTINUED FROM ABOVE**
> >
> > **Licences of other datasets: key differences**.
> > We have compiled an overview of different licences used in other dataset releases, together with background on the local jurisdictions and their customs. In short, here are the key differences between other releases and ours:
> > **Anonymity / dataset composition.** Many other datasets are from jurisdictions which anonymise their legal decisions (e.g. Korean datasets), making them legally and ethically less problematic to release.
> > **Laws.** Other papers concern jurisdictions with more permissive laws, in contrast to for instance the UK Data Protection Act / UK GDPR applying to our dataset.
> > **Our focus on legal and ethical issues.** Due to the legal and ethical background of our authorship group, these matters are part of our key focus. Others may not always be aware of the scope of the legal and ethical requirements that would apply.
> >
> >
> > **_Updates to Licence._**
> > As we are deeply committed to making our dataset more easily accessible, we took the reviewers’ suggestions very seriously and **have updated our licence to incorporate feedback and further relax some of the restrictions**. The new proposed text is now available on the website - a brief summary is given below:
> > 1. We have removed the indemnity currently required from the Dean (or equivalent) and only require a confirmation of responsibility.
> > 2. We now allow university students and researchers to have access via an application of their research supervisor.
> > 3. We emphasise that the removal of the corpus is limited to cases of legal risk.
> > With this rebalancing of access requirements, we are aiming to ensure legal and ethical use of the dataset while at the same time allowing access within these parameters. Recent months have shown that those making datasets available or using them are exposed to liability risks and we would like to avoid such risks in the spirit of legal and ethical use.
> >
> > Against this background, we will monitor the use of our dataset and will make it more easily accessible, for instance through a more permissive licence, if justified considering the usage, legal environment and ethical discussion concerning such datasets. In order to increase the uptake of the corpus by the research community, we will also advertise the corpus via various channels, e.g., the Twitter accounts of our Faculties and LinkedIn research groups.
> >
> > **Licence: code.**
> > Thank you for drawing our attention to this. Since our code does not involve any personal information, we will provide a codebase release, licensed under MIT, with the next update of the manuscript.
> >
> > **Maintenance and long-term preservation.**
> > We aim to maintain and expand the quality and quantity of the corpus. To facilitate this, we include continuous version control and quality checks, while continuously adding data to improve the corpus as we work with it. Indeed, we’ve recently obtained newer judgements up to 2023, and we are currently working on incorporating them into the corpus. Similarly, we are committed to long-term preservation, and for instance have adopted simple, standardised data formats in order to facilitate this, along with persistent storage and hosting.
> >
> > **Case outcome prediction.**
> > We agree that case outcome prediction is a very interesting topic of study, however, after internal discussion, we have concluded that properly formalising this task is highly non-trivial, because the details regarding how it is defined are crucial to what is possible.
> > In a true case outcome prediction task, the input should be an unbiased text describing the case, written by someone else prior to the judgement. The model should then be asked to predict the outcome. However, legal judgements - which, unlike other parts of the legal process, in the UK are made public - instead focus on the result, rather than on the complete process that occurs prior to the result. Thus, even with our dataset, at best we could make predictions using only part of the relevant information. In light of this, we chose not to include a case prediction task.
> > We will clarify our terminology by replacing “prediction” with “extraction” where appropriate, and modify our description to reflect that the “machine learning sentence-level prediction formalism” is used to achieve the “extraction task” (rather than a “case outcome prediction task”) to better avoid confusion.
> >
> > **Train/val/test splits: case outcome extraction.**
> > Thank you for pointing out that some of our text here was confusing. The accuracy here refers to case outcome extraction. We have updated the respective appendix to clarify this.
> >
> > **RESPONSE CONTINUED BELOW**

---

> > > ### Author Response · Authors · 2023-08-18
> > >
> > > **RESPONSE CONTINUED FROM ABOVE**
> > >
> > > **Evaluation metric: RoBERTa vs GPT.**
> > > We agree that the proposed metric, word error rate, is far from optimal, especially when comparing GPT-type models with encoder-based models. We have therefore decided to add BERTScore, BLEU and ROUGE scores.
> > > * These are GPT-3.5: [BERTScore 0.7876, BLEU 0.2169, BLEU-1 0.2568, BLEU-2 0.225, BLEU-3 0.2046, BLEU-4 0.1873, ROUGE-1 0.3078, ROUGE-2 0.2799, ROUGE-L 0.3001, ROUGE-LSum 0.2995]
> > > * These are GPT-4: [BERTScore 0.840, BLEU 0.2815, BLEU-1 0.3337, BLEU-2 0.2876, BLEU-3 0.265, BLEU-4 0.2469, ROUGE-1 0.5298, ROUGE-2 0.483, ROUGE-L 0.5172, ROUGE-Lsum 0.5167]
> > > * We will add the RoBERTa and other models when they finish computing.
> > >
> > >
> > >
> > > **Word error rate: GPT models (WER>3).**
> > > Thank you for pointing out this point of confusion. The number WER > 3 means that the GPT-models are too inclusive: they include roughly three times as much text as the original annotators. This means they have reasonable recall, but less precision. From a legal perspective, this is much preferred to the alternative where important parts are excluded, hence why we described this as good performance. We will add text clarifying this further to the manuscript.
> > >
> > >
> > > **MS Word and PDF transcription.**
> > > Thank you for raising this important point. While we have access to the original MS Word and PDF documents, we cannot legally release them. The reason for this is that, in the UK, court judgements have Crown copyright (i.e. copyright to the UK government), but are often digitised and transcribed by third-party companies which, among other things, add their company logos and similar material to the judgement PDFs. These parts of the document are potentially copyrighted by the companies, and therefore cannot be released. We have therefore taken care to ensure that only Crown copyright judgement text, which can legally be released, is contained in our dataset.
> > >
> > >
> > > **Visual cues in PDFs.**
> > > Thank you for this suggestion! We agree that visual cues are potentially important, and would be delighted to consider ways of including this. However, since this involves some degree of algorithmic design choices and additional complexity, we would prefer to defer this to future versions of the corpus.
> > >
> > > **Limitations**
> > >
> > > **Access / licence restrictions.**
> > > Please see the discussion at the very beginning of the rebuttal, where we address changes made to the licence on the basis of reviewer feedback.
> > >
> > > **Correctness**
> > >
> > > **Word Error Rate.**
> > > Please see the preceding section on “Word error rate: GPT models (WER>3)” which addresses this.
> > >
> > > **Clarity**
> > >
> > > **RoBERTa pipeline: chunking and token limits.**
> > > Thanks for pointing this out! We segmented the input documents into 512-token chunks for RoBERTa, to match its max sequence length. Additionally, for GPT-3.5 and GPT-4, we used a procedure to fit all text chunks within the 4096 token limit of the GPT-3.5-turbo model to keep data as similar as possible. Some additional details are given in Appendix F. We will describe our setups in more detail in the camera-ready version.
> > >
> > > **Joint evaluation metric.**
> > > Please see the preceding section on “Evaluation metric: RoBERTa vs GPT”, which addresses this.
> > > **Relation To Prior Work**
> > >
> > > **Extended literature review: non-licence-related.**
> > > Thank you for this comment - we agree that adding more detail on prior works can help readers find the resources they need. We have therefore conducted an additional literature review aimed at collecting the few existing legal corpora in other jurisdictions, including several references that became available after the NeurIPS submission deadline. We provide some highlights of other corpora involving case law in various jurisdictions: LexFiles (Chalkidis et al. 2023), MultiLegalPile (Niklaus et al. 2023), LEXTREME (Niklaus et al. 2023), Pile of Law (Henderson et al. 2022), (Chalkidis et al. 2020). We will add all of these and additional references to the manuscript.
> > >
> > > **Extended literature review: licensing in other jurisdictions.**
> > > We have done this -  we discuss the implications in the “Licence” section at the very beginning of our response.
> > >
> > > **Documentation**
> > >
> > > **Code Release.**
> > > In short: we will release code under MIT. Please see the response for “Licence: code.”
> > >
> > > **Licence.**
> > > Please see the extensive discussion at the beginning involving licensing.
> > >
> > > **Data Source.**
> > > In short: no. CourtCorrect has not imposed any restrictions on us with respect to release of the data. Our reasons for licensing restrictions stem from our legal and ethical considerations explained in the licence section.
> > >
> > > **RESPONSE CONTINUED BELOW**

---

> > > > ### Author Response · Authors · 2023-08-18
> > > >
> > > > **RESPONSE CONTINUED FROM ABOVE**
> > > >
> > > > **Licence: post-2003 dataset / older corporate cases**
> > > > As explained in the licence section at the beginning, we are unable to release a post-2003 dataset (or other subsets) open access for legal and ethical reasons.
> > > > Please see the licence discussion at the beginning for information about these situations.
> > > > Note also that, in light of referee feedback, we have re-evaluated and relaxed the licence restrictions, which are discussed at the beginning of our response.
> > > >
> > > > **Responsibility Statement.**
> > > > Thank you for pointing this out! This statement has now been added to our website.
> > > >
> > > > **Persistent identifier / DOI.**
> > > > This is a great idea! We have created a placeholder DOI for this dataset (we omit the number here for anonymity), and will add the final DOI to the publication once finalised.
> > > >
> > > > **Additional Feedback**
> > > >
> > > > **Topic Model evaluation.**
> > > > Thank you for bringing this to our attention. We view the topic model as an illustrative example rather than a benchmark: that is, we are interested in using the topic model to illustrate by example how the corpus can enable researchers to study how legal judgements have changed over time. For this reason, we simply picked a standard topic model, and did not focus on comparing it to alternative topic models. However, on reflection, we recognize that examining metrics such as perplexity is important, because these can potentially tell us whether or not there is a problem with the topic model. We will include this in the next version of the manuscript.
> > > >
> > > >
> > > > **Summary briefings.**
> > > > Thank you very much for this suggestion! In the UK, summary briefings are not released by the courts as part of judgments, and therefore are not part of our dataset. Instead, private legal service providers sometimes publish summaries, and only for a relatively small portion of cases. Realistically, these private service providers would not provide these briefings to us for re-publication, because their licence is on the basis of fees charged to users of the service.
> > > >
> > > > **In-context learning.**
> > > > Thank you for this suggestion! We agree that this would be interesting to explore - however, it is non-trivial due to the length of some cases relative to the context size provided by GPT models (4096 tokens), which necessitates splitting or multiple prompts or other solutions that increase the number of moving parts involved. As the primary objective of this submission is introduction of the CLC corpus and initial empirical attempts to curate the data, we believe these extensions are best suited for future work.
> > > >
> > > > **Generative modelling.**
> > > > Thanks for this suggestion - we also agree that this would be an interesting task! Indeed, there is a very large space of possible language-model-powered generative modelling use cases for this dataset. But, at this stage, we prefer to defer these to future work, and instead focus on the dataset rather than on benchmarking, as the preparatory data-quality work needs to be sufficiently complete in order to make use cases like the one you suggest possible.

---

> > > > > ### Comment · Reviewer_Z7DF · 2023-08-29
> > > > > **License restrictions**
> > > > >
> > > > > Thank you very much for your clarifications.  If, as you suggest, the CLC is comparable to medical data, there still appears to be a substantial gap between the revised CLC license and the medical dataset example referenced by NeurIPS -- the PhysioNet credentialed license https://physionet.org/about/licenses/physionet-credentialed-health-data-license-150/ mentioned in the call for Datasets and Benchmarks.  Based on that, I would consider whether the CLC can be released under a credentialized license comparable to PhysioNet, and if necessary I would consider removing the personal data from the dataset. I would also consider whether UK law permits to release an anonymized version of the CLC under an open license, like CC.  Relating to this, here are some questions/differences between the CLC and PhysioNet:
> > > > >
> > > > > * If the personal data contained in CLC necessitates the current restrictions, can the personal data be removed similarly to the de-identification of the medical data in PhysioNet? Then can the CLC be released under a license comparable to PhysioNet (assuming UK law still requires credentialized access following removal of the personal data)?
> > > > >
> > > > > * Does the personal data in the CLC add any value?  Item 3(e) in the current CLC license https://www.cst.cam.ac.uk/research/srg/projects/law says that “Research outputs, regardless of their format, must never identify natural persons, legal persons, or similar entities”.  Therefore removing the personal data seems to be desirable even under the current license as this would prevent any above-mentioned identification from happening in the first place.
> > > > >
> > > > > * The current license is still restricted to researchers at “an officially recognised university” which of course excludes researchers that do not have a university appointment. For example, the submission cites the work of the founder of the Distributed Artificial Intelligence Research Institute, but under the current license, the CLC may be unavailable to DAIR researchers without university appointments.
> > > > >
> > > > > * The license requires “ [the] Dean of the Faculty (or equivalent authority) of the researcher confirms all of the above requirements and assumes responsibility that the above requirements are met.” This requires the deans to assume *personal* liability, or at least puts them at risk of having such liability, which does not seem comparable to anything in the PhysioNet license.
> > > > >
> > > > > * The current provision in Item 3(g) is https://www.cst.cam.ac.uk/research/srg/projects/law says that “[t]he applicant agrees to immediately remove any data from the CLC as requested by one of the authors. This may involve deleting the entire CLC in case legal risks arise.“ Your response suggests that the second sentence restricts the removal of CLC data under the first sentence to legal risk cases? If so, this is unclear from the text, and in any case, there is no comparable provision in the PhysioNet license.
> > > > >
> > > > > I have a few additional comments:
> > > > >
> > > > > * The UK open justice license excludes “computational analysis of the Information” — see “Exclusions” in https://caselaw.nationalarchives.gov.uk/open-justice-licence Would you be able to clarify why this does not apply to the dataset?
> > > > >
> > > > > * The required author statement mentioned in https://nips.cc/Conferences/2023/CallForDatasetsBenchmarks under Submission Instruction is that the authors bear all responsibility in case of violation of rights appears to be missing. The instructions require this statement to be included in “supplemental materials (as a separate PDF)”, i.e, part of the published work rather than the website (but I didn’t find it on the CLC website either).
> > > > >
> > > > > * Given the importance and bespoke nature of the license, will its principal terms and restrictions be explained in the main paper or at least in the supplemental materials (at the moment supplemental materials only refer to the website regarding licensing)?

---

> > > > > > ### Author Response · Authors · 2023-08-30
> > > > > >
> > > > > > Thank you very much for your reply, and, again, for your great suggestions! Let us comment on each of your remaining points below.
> > > > > >
> > > > > > **De-identification of personal data (questions)**.
> > > > > > We agree that this would be ideal, but given the nature of the UK’s legal system, this would require us to **manually remove the parties’ names, as well as names of judges, solicitors, barristers, witnesses, experts, and other parties in the cases**. This would also require us to remove parties’ names from _case names_, making it impossible to reference or cross-link many cases, significantly decreasing the value of the dataset. De-identification would, realistically speaking, require us to manually read and edit a very large number (200k+) of cases by hand, because in the **UK system there is no consistency in how people’s names are referenced**. This is in contrast with other systems, which are highly standardised. This means that automated solutions would sacrifice quality and leave mistakes. In this respect, UK legal data is completely different from medical data, where anonymisation is much easier to achieve to a certain degree, by simply removing patient names and similar items from forms. De-identification is therefore unfortunately not feasible with the resources our team has access to.
> > > > > >
> > > > > > **Value of personal data (questions)**.
> > > > > > In the UK legal system, _case names_ are often considered personal information, because they contain the names of the parties. These names are then used by judges to reference other cases, including in various non-standardised ways that make it difficult to substitute them in any automated manner. Removing case names, which would be necessary to anonymise the dataset, would therefore make it very difficult to cross-reference cases, and strip away most references, resulting in the loss of information which is critically important to legal research.
> > > > > >
> > > > > > **Licence: non-university researchers (questions)**.
> > > > > > Thanks for this suggestion! We’ve looked at this again, and, following your suggestion, clarified that the relevant UK research exemptions are not limited to universities. **We will therefore expand our licence to also include researchers at non-university institutions, with language that mirrors UK data protection law**. We will add this to the website shortly.
> > > > > >
> > > > > > **Licence: responsibility statement (questions)**.
> > > > > > Under our licence, the Dean (or equivalent) does not take on _personal_ liability as an individual, but rather attests that the university (or equivalent) as an organisation accepts _professional_ responsibility to ensure the legal compliance of the relevant research team. We will clarify the text to ensure this is not ambiguous. Please note that, in legal compliance and ethics, one needs to set up governance mechanisms that ensure the compliance of the rules. We believe that this mechanism is necessary at this time in order to provide an additional check to ensure the research performed with this dataset is legal and ethical.
> > > > > >
> > > > > > **Clarification of 3(g) (questions)**.
> > > > > > Thanks for spotting this. We are happy to clarify that the second sentence also applies to the first sentence.
> > > > > >
> > > > > > **UK Open Justice License (additional comments)**.
> > > > > > Thanks for this question. Please be advised that the UK Open Justice License - which is more restrictive than ours - does not apply to us. This is because we have not obtained our dataset under that licence. Since we believe that enabling computational analysis - which is prohibited under the Open Justice License - is a critically important contribution of our research, we would rather apply our own, less restrictive licence.
> > > > > >
> > > > > > **Author Statement of Responsibility (additional comments)**.
> > > > > > Thank you for bringing this up. The NeurIPS instructions are ambiguous as to whether this statement takes the form of an OpenReview checkbox, or text that we need to add in the supplementary material. We will clarify this with the conference organisers, and will add all required statements.
> > > > > >
> > > > > > **Explanation of Licence Terms in Main Paper (additional comments)**.
> > > > > > Thank you for this. We will expand on the legal and ethical section of the main paper and appendix to incorporate the feedback above. This will include additional text on what specific forms of personal information are contained in the data, and what specific UK legal exemptions we rely on to enable legal access.

---

### Official Review · Reviewer_rSsA · 2023-07-26
**Good resource, good evaluation on case extraction, needs corrections, and more analysis.**

**Rating:** 6
**Confidence:** 5
**Clarity:** The paper is quite clearly written.

**Strengths:**

A large resource which should be utilized for legal AI research.
Case outcome extraction task is well-motivated
Experimental results show data usability, and this data can have a considerable impact.
Strong ethical considerations; have been described well.

**Additional Feedback:**

None

**Correctness:**

The claims made are largely correct, except there is missing analysis pointed out above.


**Documentation:**

The documentation seems to be good from the appendix.

**Ethics:**

Strong ethical considerations required for such a resource, as reflected in the paper.

**Limitations:**

Given the licensing details for this data, its use for research purposes can be encouraged.

**Opportunities For Improvement:**

- Line 309: Both RoBERTa models do not obtain >99% accuracy according to Table 1.
- How do the authors arrive at the prompt used? Were there candidate prompts first? Details are not clear.
       - Prompt not provided for the zero-shot task on LLMs, a sample prompt should be in the main draft, the rest of them can be in the
         appendix.
- The authors should consider providing an error analysis of the samples, which results in low F1 scores or depleting accuracy.


**Relation To Prior Work:**

It's a novel resource and discusses such prior corpora but not in sufficient detail and without any comparison with any such existing corpus.

**Summary And Contributions:**

This paper describes the curation of the Cambridge Law Corpus (CLC) for legal AI research consisting of more than 250k court cases along with their meta-data. The authors perform analysis on various corpus domains like Immigration, Employment and so on to show how the domain-specific proportion of this data spans over various year spans, using topic modelling. Further, they perform experiments with a 638 case sample to predict the outcome of a case. The authors employ a mask-filling approach with the RoBERTa model, and perform zero-shot experiments with two GPT models to show how case outcomes can be modelled. The experiments show a decent result with high accuracy but considerably lower F1 score for the end-to-end RoBERTa model.

Strong contribution in terms of a resource but please see weaknesses below and provide corrections accordingly.

---

> ### Author Response · Authors · 2023-08-16
>
> We thank you for your time and insightful comments! We are delighted that you thought that our data “can have considerable impact” and that our example tasks are “well-motivated”. Below we address the questions raised:
>
> **Opportunities For Improvement**
>
> **RoBERTa model accuracy.**
> Thanks for identifying this - we accidentally missed this line when updating the paper with the finalised results. The correct values are given in Table 1, and are 99.7% for end-to-end RoBERTa and 73.9% for pipeline-based RoBERTa. The line will instead read “Both RoBERTa-based models obtain relatively-high accuracies, but comparatively-low F1 scores, with end-to-end RoBERTa favouring a tradeoff more in the direction of higher accuracy.” We have now fixed this.
>
> **Construction of prompts.**
> Thank you for this suggestion - our prompt creation process is at present briefly described in the appendix, and in short involves starting from a hand-written initial prompt, asking ChatGPT how to improve the prompt, testing out the new prompt, and repeating this process recursively. In retrospect, we recognize the importance of this, and will add a short description to the main manuscript, and expand the amount of detail given in the appendix.
>
>
>
> **Error analysis of samples, accuracy / F1 scores.**
> We provide some discussion in lines 309-315 and in Appendix C. In summary, the main reason for the accuracy and F1 scores observed appears to be that case outcomes are described by very few tokens in relation to the length of the case, which leads to a **large class imbalance in the classification problem**. This means that a model can achieve high accuracy (but low F1 score) by simply classifying all tokens as “not case outcome”.
>
>
>
> **Licensing details / research purposes.**
> Thank you for this important question. We reached our licensing arrangement by balancing two competing factors: (1) making the corpus as widely available as possible while (2) at the same time respecting the legal and ethical limitations set out in the paper and the data questionnaire.
> * Against this background, the corpus is currently restricted to be used for research, and ethical assurances such as an ethics clearance have to be submitted for access. This mirrors similar restrictions used in other works, for instance medical datasets and those with personal information, that have previously been published within the NeurIPS dataset track.
> * As we are deeply committed to making our dataset more easily accessible, we took the reviewers’ suggestions very seriously and **have updated our licence to incorporate feedback and further relax some of the restrictions**. The new proposed text is now available on the website - a brief summary is given below:
> * 1. We have removed the indemnity currently required from the Dean (or equivalent) and only require a confirmation of responsibility.
> * 2. We now allow university students and researchers to have access via an application of their research supervisor.
> * 3. We emphasise that the removal of the corpus is limited to cases of legal risk.
> * Against this background, we will monitor the use of our dataset and will make it more easily accessible, for instance through a more permissive license, if justified considering the usage, legal environment and ethical discussion concerning such datasets.
> * In order to increase the uptake of the corpus by the research community, we will also advertise the corpus via various channels, e.g., the Twitter accounts of our Faculties and LinkedIn research groups.
>
> **Relation To Prior Work:**
>
> **Additional Literature Review**
> Thank you for this comment - we agree that adding more detail on prior works can help readers find the resources they need. We have therefore conducted an additional literature review aimed at collecting the few existing legal corpora in other jurisdictions, including several references that became available after the NeurIPS submission deadline. We provide some highlights of other corpora involving case law in various jurisdictions: LexFiles (Chalkidis et al. 2023), MultiLegalPile (Niklaus et al. 2023), LEXTREME (Niklaus et al. 2023), Pile of Law (Henderson et al. 2022), (Chalkidis et al. 2020). We will add all of these and additional references to the manuscript.
>
> **Flag For Ethics Review:**
>
> Thank you for flagging this! Given our subject matter, if not flagged by a reviewer, we would have self-flagged we have carefully taken legal and ethical considerations into account throughout our work, but also believe given the nature of the data that a comprehensive review as part of the submission process is necessary and appropriate.

---

### Author Response · Authors · 2023-08-29
**Summary of Responses**

We thank all reviewers taking the time to read our paper and for the insightful and helpful comments offered! We are pleased the reviewers found the introduction of a corpus of curated UK cases to be timely, relevant, and important for NLP research.

The most pressing concerns are as follows:

**Licence - legal and ethical side.** There is some disagreement among reviewers: Reviewer Z7DF asks for a more permissive open access licence, while Reviewer Hrgy praises our work for careful consideration of legal and ethical issues. _We believe this was the single biggest concern raised._
* **We have adjusted our licence to reflect reviewer feedback** to better balance legal concerns while maximising access. Please note that due to the specific combination of factors in the UK, including non-anonymised court decisions combined with strong privacy protections (such as the Data Protection Act / UK GDPR), we believe that **releasing this dataset under a full open access licence (such as CC) would potentially be illegal in the UK** and risk litigation. Using our team’s legal and ethical expertise, we have strived to create a licence which is **as permissive as possible while limiting risks**, by allowing us to take advantage of, for instance,  research exemptions to the UK Data Protection Act / UK GDPR to ensure legality. Our resulting licence and restrictions are similar to those used in medical and other sensitive datasets published previously via the NeurIPS Dataset Track.

**Benchmarking and evaluation.** The reviewers (rSsA, Z7DF, Sw7Y) offered a number of suggestions for better evaluation of our examples, such as extra metrics.
* **We have computed these**, posted the results in our rebuttal, and will be adding them to the paper.

**Prompting.** The reviewers (rSsa, Hryg) wanted more explanation of the process we used to arrive at a suitable prompt for the GPT models.
* **We have written an additional summary** of our process in the responses, and will add the information in this response to the appendix.

**Data quality information.** The reviewers (JUfX, Hryg, Sw7Y) wanted to know more about our data quality assurance process, including in particular the OCR error rate, asked for additional sample cases, and asked for additional details about how we plan to audit changes and corrections made to the corpus.
* **We have done this**, and have provided a quick estimate of the OCR error rate based on sampling random documents, provided more sample cases, and added additional details on quality assurance in our responses, and we will add more about these points to the manuscript.

---

### Decision · Program_Chairs · 2023-09-22

**Decision:**

Accept (Poster)

**Comment:**

The paper introduces a new corpus for legal AI research, which is also tested with various language models. All reviewers and I agree that this an interesting paper which presents a potentially very relevant and useful dataset. The very low score of reviewer JUfX is not justified adequately, since its review is absolutely uninformative. The authors also satisfactorily addressed all the points raised by the reviewers in their rebuttal.